# Molecular epidemiology and antimicrobial susceptibility of diarrheagenic *Escherichia coli* isolated from children under age five with and without diarrhea in Central Ethiopia

Tizazu Zenebe Zelelie[1,2,3]*, Tadesse Eguale[4,5], Berhanu Yitayew[1], Dessalegn Abeje[3], Ashenafi Alemu[3], Aminu Seman[2], Jana Jass[6], Adane Mihret[2,3], Tamrat Abebe[2]

1 Department of Medical Laboratory Science, Debre Berhan University, Debre Birhan, Ethiopia, 2 Department of Microbiology, Immunology and Parasitology, Addis Ababa University, Addis Ababa, Ethiopia, 3 Armeur Hansen Research Institue (AHRI), Addis Ababa, Ethiopia, 4 Aklilu Lemma Institute of Pathobiology, Addis Ababa University, Addis Ababa, Ethiopia, 5 Ohio State University Global One Health LLC, Addis Ababa, Ethiopia, 6 The Life Science Centre—Biology, School of Science and Technology, Orebro University, Örebro, Sweden

* tizazuzenebe@yahoo.com, zelelietizazu@gmail.com

**Data Availability Statement:** All relevant data are within the paper and its Supporting Information files.

## Abstract

### Background

Diarrhea is a serious health problem in children, with the highest mortality rate in sub-Saharan Africa. Diarrheagenic *Escherichia coli* (DEC) is among the major bacterial causes of diarrhea in children under age five. The present study aims to determine molecular epidemiology and antimicrobial resistance profiles of DEC and identify contributing factors for acquisition among children under age five in Central Ethiopia.

### Methods

A health facility-centered cross-sectional study was conducted in Addis Ababa and Debre Berhan, Ethiopia, from December 2020 to August 2021. A total of 476 specimens, 391 from diarrheic and 85 from non-diarrheic children under age five were collected. Bacterial isolation and identification, antimicrobial susceptibility, and pathotype determination using polymerase chain reaction (PCR) were done.

### Results

Of the 476 specimens analyzed, 89.9% (428/476) were positive for *E. coli*, of which 183 were positive for one or more genes coding DEC pathotypes. The overall prevalence of the DEC pathotype was 38.2% (183/476). The predominant DEC pathotype was enteroaggregative *E. coli* (EAEC) (41.5%, 76/183), followed by enterotoxigenic *E. coli* (21.3%, 39/183), enteropathogenic *E. coli* (15.3%, 28/183), enteroinvasive *E. coli* (12.6%, 23/183), hybrid strains (7.1%, 13/183), Shiga toxin-producing *E. coli* (1.6%, 3/183), and diffusely-adherent *E. coli* (0.6%, 1/183). DEC was detected in 40.7% (159/391) of diarrheic and 28.2% (24/85) in non-diarrheic children (p = 0.020). The majority of the DEC pathotypes were resistant to

**Funding:** The authors did not receive specific funding for this work.

**Competing interests:** The authors have declared that no competing interests exist.

**Abbreviations:** AMR, Antimicrobial resistance; CLSI, Clinical and Laboratory Standards Institute guidelines; DAEC, Diffusely adherent *E. coli*; DEC, Diarrheagenic *Escherichia coli*; DNA, Deoxyribonucleic acid; EAEC, Enteroaggregative *E. coli*; EIEC, Enteroinvasive *E. coli*; EPEC, Enteropathogenic *E. coli*; ESBLs, Extended spectrum beta lactamase; MDR, Multi-drug-resistant; PCR, Polymerase Chain Reaction; STEC, Shiga toxin producing *E. coli*; ETEC, Enterotoxigenic *E. coli*; ATCC, American Type Culture Collection; SPSS, Statistical Package for Social Sciences; CCUG, Culture Collection University of Gothenburg; EDTA, Ethylenediamine tetraacetic acid.

ampicillin (95.1%, 174/183) and tetracycline (91.3%, 167/183). A higher rate of resistance to trimethoprim-sulfamethoxazole (58%, 44/76), ciprofloxacin (22%, 17/76), ceftazidime and cefotaxime (20%, 15/76) was seen among EAEC pathotypes. Multidrug resistance (MDR) was detected in 43.2% (79/183) of the pathotypes, whereas extended spectrum ß-lactamase and carbapenemase producers were 16.4% (30/183) and 2.2% (4/183), respectively.

## Conclusion

All six common DEC pathotypes that have the potential to cause severe diarrheal outbreaks were found in children in the study area; the dominant one being EAEC with a high rate of MDR.

## Introduction

Diarrheal diseases in children under age five contribute to high mortality rates in sub-Saharan Africa and Central and Southern Asia [1]. Pneumonia and diarrhea cause 140 death per hour and 3400 death per day for children before celebrating their fifth birthday [2]. Ethiopia ranks fifth among the 15 pneumonia and diarrhea high-burden countries in deaths (due to pneumonia and diarrhea) of children under age five [2]. And diarrhea is the second leading cause (after respiratory infections) of death in children aged 1–4 years in Ethiopia [3]. A systematic review and meta-analysis conducted in Ethiopia reported a 22% prevalence of diarrhea among children under age five [4].

The main etiological agents of diarrheal diseases include viruses (i.e. rotavirus) and enteric bacteria (*Shigella*, *Vibrio cholera*, *Escherichia coli*, and *Salmonella*) [5]. Among the bacterial agents, diarrheagenic *Escherichia coli* (DEC) is the major causal agent of diarrhea in children under age five [6] and is responsible for severe diseases and outbreaks [7]. Specific combinations of virulence traits group DEC into six common pathotypes, including enterotoxigenic *E. coli* (ETEC), enteropathogenic *E. coli* (EPEC), Shiga-toxin producing *E. coli* (STEC) /enterohemorrhagic *E. coli* (EHEC), enteroinvasive *E. coli* (EIEC), enteroaggregative *E. coli* (EAEC), and diffusely adherent *E. coli* (DAEC) [6]. ETEC, a major cause of traveler's diarrhea, produces either a heat-labile (LT) or a heat-stable (ST) enterotoxin, and a set of colonization factors [6]. EPEC is an attaching and effacing (A/E) pathogen able to form distinctive lesions (A/E lesions) on the surface of intestinal epithelial cells [6]. Based on the presence or absence of *E. coli* adherence factor plasmid (pEAF), EPEC strains are classified into typical EPEC (tEPEC) that has the pEAF, and atypical EPEC (aEPEC) that lacks the pEAF [6]. The plasmid (pEAF) carried a gene called *bfp* that encodes bundle-forming pili (BFP), an important virulence factor of the tEPEC strains. Both tEPEC and aEPEC produce a potent adherence factor (*eae)* called intimin which is important for the intimate attachment of EPEC to intestinal cells during A/E lesion formation [6]. STEC is a foodborne and zoonotic pathogen, and it causes non-bloody diarrhea, bloody diarrhea, hemorrhagic colitis, and haemolytic uremic syndrome [6]. STEC is known for its potent toxin called Shiga toxin, which is encoded by *stx* genes (*stx1* and *stx2*) [6]. EIEC is a facultative intracellular pathogen that produces specific virulence factors involved in invasion including invasion plasmid encoding genes (*ipaH*) and transcription activator (*virF*) [6]. The virF is a DNA-binding protein (also called master regulator) that controls the expression of virulence factors (e.g. T3SS and effector proteins) by regulating *virB* (another transcriptional regulator) and *icsA* genes [6, 8]. *VirB* and *IcsA* (VirG) are required to the full expression of the invasion program by EIEC. EAEC causes persistent diarrhea in children (endemic) and

traveler's diarrhea. EAEC produces specific virulence factors including a transcriptional activator gene (*aggR*) and/or the enteroaggregative heat-stable toxin-1 (EAST-1, *astA*). DAEC is known by its fimbrial adhesion genes (*daaD* and/or *daaE*). Recently hybrid pathotypes have been reported that have virulence factors associated with more than one pathotype [9].

The genomic plasticity of *E. coli* results in the emergence of hybrid virulent and multi-drug-resistant (MDR) strains [10] that have the potential to cause severe clinical diseases and outbreaks. Hybrid DEC pathotype strains are increasingly reported in different countries of the world. Among the hybrid strains reported include EPEC/ETEC in India [11], STEC/EAEC in Germany [12] and Indonesia [13], EPEC/STEC [14] and EPEC/EAEC [15] in Brazil, and STEC/ETEC in Sweden [16]. Outbreaks of diarrheal diseases due to different DEC pathotypes reported in countries include Germany in 2011 [17], Japan in 2016 [18] and 2020 [19], Nottingham, UK in 2014 [20], South Korea in 2018 [21], and the United Kingdom in 2020 [22]. In addition, outbreaks due to different *E. coli* strains that occurred annually are available in the Centers for Disease Control and Prevention outbreak report (https://www.cdc.gov/ecoli/2022-outbreaks.html). Antimicrobial resistance (AMR) among enteric pathogens is also a serious concern with increased mortality and high public health costs [23]. High frequencies of MDR DEC pathotypes in children are reported in India, 41% [24], and China, 67% [25]. From MDR bacterial strains, the extended-spectrum ß-lactamase (ESBL) and carbapenemase-producing DEC pathotypes are among the emerging pathogens globally [24]. Nowadays, there is a serious threat due to ESBL-producing Enterobacteriaceae and an urgent threat due to Carbapenem-resistant *E. coli* [26]. The occurrence of hybrid [11–14], outbreak-causing [17, 18, 20], and resistant [24, 25] DEC strains in different areas of the world could reveal the presence of a threat for the possible occurrence of DEC-caused severe diseases or outbreaks. Epidemiological data of DEC will initiate public health personnel to strengthen clinical laboratory diagnostic capacity and establish an active surveillance program that enables identifying and tracking bacterial outbreaks and severe diarrheal diseases. Thus, understanding the distribution pattern of enteric pathogens within the community will help control or prevent fatal outbreaks.

Poor hygiene, sanitation, and lack of access to clean water supplies are associated with diarrhea in developing countries [27]. Climate variability affects the availability of clean water and effective sanitation, and this mediates the causes of diarrheal diseases [28]. A systematic review and meta-analysis found an 8% increase in the incidence of diarrheagenic *E. coli* (DEC) for each 1˚C increase in mean monthly temperature [29]. Ethiopia is vulnerable to increasing temperatures and climate change [30] which exacerbates the effects of poor sanitation [31], interrupted water supply [32], and unimproved drinking water sources [31]. DEC strains are transmitted predominantly through faecal-oral contact via contaminated food and water [6]. The presence of these potential risk factors makes Ethiopia more vulnerable to DEC, with the largest burden among children under age five.

The molecular epidemiology of DEC pathotypes in children under age five in Ethiopia is not well known. The present study aims to determine the occurrence of DEC pathotypes and their AMR profile in Addis Ababa and Debre Berhan, Ethiopia, in children under age five using a health-care facility-based study. Furthermore, risk factors contributing to DEC prevalence were also identified.

## Materials and methods

### Study area and participants

A health-care based cross-sectional study was conducted in Addis Ababa and Debre Berhan, Ethiopia, from December 2020 to August 2021. Addis Ababa is the capital city of Ethiopia and, as the headquarters of the African Union; it is the focal point of international activities. The

recent population estimate for Addis Ababa is 5.2 million (https://worldpopulationreview.com/world-cities/addis-ababa-population). Debre Berhan is located about 130 km from Addis Ababa in the North Shoa Zone of the Amhara regional state, with a population of nearly 95,000 (https://edaethiopia.org/index.php/program-offices/8-debrebirhan).

Three health-care centers from Addis Ababa and two from Debre Berhan were randomly selected to participate in the study. Study participants were enrolled based on inclusion and exclusion criteria. The inclusion criteria were children under age five, who did not receive antibiotics in the past three weeks, and children with diarrhea. Children with the age of five or older age, and who received antibiotics treatment were the exclusion criteria. The same inclusion criteria (except diarrheic) and those who attended the health facility for causes other than diarrhea were used to include non-diarrheic children in the study. Prior antibiotic use can affect pathogen distribution [33]. Although the exact time for antibiotics to stay in the body is varied (due to many factors), some antibiotics stay a short time (24hrs) [34], and others, such as azithromycin [35], stay a longer time (more than two weeks). For this reason, only those who had no antibiotic exposure in the past three weeks were included in the study. Socio-demographic, clinical features and other factors were collected from the parents/guardians using standardized structured questionnaires. The socio-demographic data included age, sex, and family income. The clinical features included initiation of illness, duration of diarrhea, stool frequency, type of diarrhea, dehydration status, fever, vomiting, nausea, thirst, abdominal distension, and history of previous treatment. In addition, feeding practices, child care, access to animals, and sources of drinking water were also included. The questionnaire used for the present study is available in supplementary file (S1 File). The study participant's recruitment (data collection) was started on December 2020 and lasted on August 2021 (9 months period).

## Specimen collection

Training has been given to data collectors on the aseptic procedures of the stool sample collection from children to avoid contamination of the stool samples (with urine, soil, or water). How to use the sterile collection materials (transport media, plastic wrap, and wooden sticks) and the critical steps in taking and transferring the stool samples were given during the training. Briefly, stool samples were collected by placing plastic wrap over the rim of a potty or using a disposable diaper with the plastic side next to the skin. Feces containing blood, mucus, or pus were transferred to Cairy-Blair transport media using two small sterile wooden sticks and transported to the Microbiology Laboratory of Tikur Anbessa Specialized Hospital for the Addis Ababa site and Debre Berhan Referral Hospital for the Debre Berhan site using cold packs. Then, bacterial isolation and identification, and antimicrobial susceptibility testing were done. Conventional Polymerase Chain Reaction (PCR) assay for the detection of DEC-specific virulence genes was performed at Armauer Hansen Research Institute.

## Bacterial isolation and identification

Faecal suspension was prepared by taking the wooded sticks (cotton swabs) from the transport media containing the stool sample and rinsed thoroughly in 1 ml of saline. For the liquid stool sample, saline was not used. Then, a loopful of faecal suspension (liquid stool sample) was inoculated onto MacConkey agar (Oxoid Ltd, Basingstoke, UK), and incubated at 37˚C for 18–24 hours. *E. coli* isolates were identified using conventional biochemical tests including oxidase, triple sugar iron agar, urease, lysine decarboxylase, motility, indole, Simon's citrate, and hydrogen sulphide [36]. Confirmed *E. coli* isolates from each cultured plate were stored at -80˚C in brain heart infusion broth containing 16% (v/v) glycerol.

## Antimicrobial susceptibility testing

Antimicrobial susceptibility tests for all DEC pathotypes were assessed using the disc diffusion method (Kirby–Bauer method) according to Clinical and Laboratory Standards Institute guidelines (CLSI) [37]. Antibiotics for the study were selected based on CLSI guideline for enterobacterales. The antibiotics tested were ampicillin (AM 10 μg; oxoid), ceftazidime (CAZ 30 μg, oxoid), cefotaxime (CTX 30 μg; oxoid), ertapenem (ETP 10 μg; oxoid), meropenem (MEM 10 μg; oxoid), amoxicillin-clavulanate (AMC 20/10 μg; oxoid), gentamycin (GM 10 μg; oxoid), tetracycline (T 30 μg; oxoid), trimethoprim-sulfamethoxazole (SXT 1.25/23.75 μg; oxoid), ciprofloxacin (CIP 5 μg; oxoid), chloramphenicol (C 30 μg; oxoid), and cefepime (FEP 30 μg; oxoid). The results were interpreted using CLSI guidelines [37]. *E. coli* ATCC (American Type Culture Collection) 25922 was used as quality control strain for the antimicrobial susceptibility testing. MDR was defined when the isolate was non-susceptible to at least one agent (from each class) in three or more antimicrobial classes [38].

## Phenotypic detection of ESBLs and Carbapenemases production

DEC strains that were resistant at least to cefotaxime (30 μg) or ceftazidime (30 μg) in the screening test were selected. The ESBL production was confirmed using the combination disk method [37]. Briefly, the combination disk method was done on MHA by using ceftazidime (CAZ) and cefotaxime (CTX) alone and with ceftazidime + clavulanic acid (CAZ/CLA) and cefotaxime + clavulanic acid (CTX/CLA) as recommended by CLSI guidelines [37]. The increase in zone size diameter by ≥ 5 mm for CTX/CLA and CAZ/CLA, when compared with that CTX and CAZ alone, was confirmed as the presence of ESBL. *K. pneumoniae* ATCC 700603 (ESBL positive) and *E. coli* ATCC 25922 (ESBL negative) were used for quality control. The production of carbapenemase was confirmed using the modified carbapenem inactivation method [37]. *K. pneumonia* ATCC BAA-1705 (carbapenemase positive) and *K. pneumonia* ATCC BAA-1706 (carbapenemase negative) were used for quality control. Briefly, a loopful (by using a 1 μL size loop) of the colony of the test isolates from overnight blood agar plate suspended in 2 mL of nutrient broth (Oxoid UK) and 10 μg of meropenem disk immersed in the nutrient broth (fully immersion). The tubes were incubated at 37˚C in ambient air without agitation for 4 h ± 15 min. Subsequently, the meropenem disks were removed from the broth using an inoculation loop. Then, the disks were placed on Mueller-Hinton agar plates (Oxoid, UK) freshly inoculated with a 0.5 McFarland suspension of *Escherichia coli* ATCC 25922. The results from the ESBL confirmation and carbapenemase production were interpreted according to CLSI guidelines [37].

## Detection of DEC virulence genes

Genomic DNA (deoxyribonucleic acid) extraction was performed using the boiling method as described by Rúgeles *et al.* [39]. Briefly, an overnight liquid culture suspension of DEC isolates was boiled at 94˚C for 10 min in a dry block incubator (Thermo Fisher Scientific, California) and placed in a freezer at −20˚C for 10 minutes, then placed at room temperature for one minute and centrifuged at 14,000 g for 5 min. Then, 100 μL of the supernatant was transferred into a nuclease-free Eppendorf tube and stored at −20˚C until use.

Identification of DEC was performed using multiplex PCR following a previously described procedure [39], with slight modification. Briefly, the detection of DEC virulence genes was performed in three separate PCR reactions (multiplex) using the primer sequences shown in Table 1. The three PCR reactions were set based on the base pair sizes (Table 1) of the target genes. PCR reaction 1 contained primer mix 1 (M-1) for the detection of EPEC and STEC targeting virulence genes *bfp*, *eae*, and *stx*. PCR reaction 2 contained primer mix 2 (M-2) for

**Table 1. Target genes for PCR amplification, primers and their characteristics.**

| PCR reaction mix | Strains | Target genes | Primer sequence | Amplicon size (bp) | Ref. |
|---|---|---|---|---|---|
| M-1 | EPEC/ STEC | *eae* | F: 5′-CTGAACGGCGATTACGCGAA-3′<br>R: 5′-CGAGACGATACGATCCAG-3′ | 917 | [39] |
| | EPEC (typical) | *bfp* | F: 5′-AATGGTGCTTGCGCTTGCTGC-3′<br>R: 5′-GCCGCTTTATCCAACCTGGTA-3′ | 326 | |
| | STEC/ EHEC | *stx* | F:5′-GAGCGAAATAATTTATATGTG-3′<br>R: 5′-TGATGATGGCAATTCAGTAT-3′ | 518 | |
| M-2 | ETEC | *lt* | F: 5′-GCACACGGAGCTCCTCAGTC-3′<br>R: 5′-TCCTTCATCCTTTCAATGGCTTT-3′ | 218 | |
| | | *st* | F: 5′-GCTAAACCAGTAGAG(C)TCTTCAAAA-3′<br>R: 5′-CCCGGTACAG(A)GCAGGATTACAACA-3′ | 147 | |
| | EIEC | *virF* | F: 5′-AGCTCAGGCAATGAAACTTTGAC-3′<br>R: 5′-TGGGCTTGATATTCCGATAAGTC-3′ | 618 | |
| | | *ipaH* | F: 5′-CTCGGCACGTTTTAATAGTCTGG-3′<br>R: 5′-GTGGAGAGCTGAAGTTTCTCTGC-3′ | 993 | |
| M-3 | EAEC | *aggR* | F: 5′-GTATACACAAAAGAAGGAAGC-3′<br>R: 5′-ACAGAATCGTCAGCATCAGC-3′ | 254 | |
| | | *astA* | F:5′-CCATCAACACAGTATATCCGA-3′<br>R: 5′-GGTCGCGAGTGACGGCTTTGT-3′ | 111 | [40] |
| | DAEC | *daaF* | F: 5′-GAACGTTGGTTAATGTGGGGTAA-3′<br>R: 5′-TATTCACCGGTCGGTTATCAGT-3′ | 542 | [39] |

bp, base pair; PCR = Polymerase Chain Reaction; EPEC, Enteropathogenic *Escherichia coli*; ETEC, Enterotoxigenic *Escherichia coli*; EIEC, Enteroinvasive *Escherichia coli*; EAEC, Enteroaggregative *Escherichia coli*; STEC, Shiga-toxin producing *Escherichia coli*; DEAE, Diffusely adherent *Escherichia coli*

ETEC and EIEC targeting virulence genes *lt*, *st*, *virF*, and *ipaH*. PCR reaction *3* contained primer mix 3 (M-3) for the detection of EAEC and DAEC targeting virulence genes *aggR*, *astA*, and *daaF*. Multiplex PCR reactions 1 and 3 were carried out with 20 μl reaction mixture containing 10 μl Platinum™ II Hot-Start PCR Master Mix (2X) (ThermoFisher Scientific), 1.2 μl of the forward primer mix (0.4 μl each primer), 1.2 μl of reverse primer mix (0.4 μl each primer), 1 μl of template DNA, and 6.6 μl of molecular grade water. Multiplex PCR reaction 2 was also carried out with 20 μl reaction mixture containing 10 μl Platinum™ II Hot-Start PCR Master Mix (2X) (ThermoFisher Scientific), 1.6 μl of the forward primer mix (0.4 μl each primer), 1.6 μl of reverse primer mix (0.4 μl each primer), 1 μl of template DNA, and 5.6 μl of molecular grade water. The PCR thermal conditions were set with an initial denaturation of 94°C for 2 min, followed by 35 cycles of 92°C for 30 sec, annealing at 60°C for 30 sec, extension at 72°C for 30 sec, and a final extension at 72°C for 5 min in a PCR machine (Biometra TRIO Thermal Cycler, Analytik Jena). PCR products were separated on 1.7% (w/v) agarose gel in Tris Borate EDTA (Ethylenediamine tetraacetic acid) buffer (pH 8.2) stained with ethidium bromide (10 μg/ml) and visualized with UV transilluminator system (Bio-Rad). DEC pathotypes (whole genome sequenced strains) confirmed positive for the target virulence gene(s) from previous works used as the positive control. The CCUG (Culture Collection University of Gothenburg) 24T *E. coli* strain confirmed negative to the target gene (s) was used as a negative control. All isolates generating positive results in multiplex PCR mix 1 or mix 2 or mix 3 were retested by a confirmatory singleplex PCR following DNA re-isolation.

## Data analysis

Data were entered into an Excel spreadsheet and cross-checked for correctness before analysis. Data analysis was made by transforming the data from Excel into SPSS Statistical Package for Social Sciences) software program version 20. Descriptive statistics were performed using the

program, and the difference between variables was computed by using Chi-Square statistics. Determinants of DEC acquisition among children under age five were identified using a binary logistic regression (Bivariate and multivariate analyses). Independent variables for the final model (multivariate logistic regression) were identified using a bivariate logistic regression model with p < 0.25. The independent effect of each independent variable on the study variable, with a significance value of p < 0.05 was determined using multivariate logistic regression. Model fitness was checked by the Hosmer and Lemeshow goodness-of-fit (p = 0.348). Multi-co-linearity of the independent variables was tested using the Variance Inflation Factor (VIF) and the Tolerance tests. Crude odds ratio (COR) and adjusted odds ratio (AOR) were used to present the results.

### Ethics approval

The study was approved by the Institutional Review Board of the College of Health Sciences, Addis Ababa University (protocol number: 025/20/DMIP) and the Ethiopian National Research Ethics Review Committee (Ref.No. RED/1.14/9428/21). Informed verbal assent and written consent were obtained from parents/guardians at the time of data collection.

## Results

### Socio-demographic characteristics of the study population

A total of 476 children under age five were included in this study, where 277 (58.2%) were from Addis Ababa and 199 (41.8%) were from Debre Berhan. Of these, 274 (57.6%) were male, and 202 (42.4%) were female (Table 2). The majority of the children were 24–59 months (60.5%, 288/476). Of the 476 participants, 391 (82%) were diarrheic and 85 (18%) were non-diarrheic. During the winter and spring (the dry season), 326 samples (273 diarrheic and 53 non-diarrheic children) were collected. The remaining 150 samples (118 diarrheic and 32 non-diarrheic children) were collected in the summer (the rainy season). The family income was categorized according to the previous study done by Fekadu and Lemma [41] and the Ethiopian birr was converted to US dollars at an exchange rate during collection (Table 2). The number of children in the family was one in 40.3% (192/476), two in 38.9% (185/476), and three in 20.8% (99/476). The clinical presentation of children under age five positive for DEC is presented in S1 Table.

### Epidemiology of DEC and associated factors

Of the 476 specimens analyzed by culture, 89.9% (428/476) were positive for *E. coli*. All 428 *E. coli* isolates were tested by PCR, of which 183 were positive for the DEC pathotype. The overall prevalence of DEC in the present study was 38.4% (183/476). Of the total DEC pathotypes, 58.5% (107/183) were from Addis Ababa, and the remaining 41.5% (76/183) were from Debre Berhan (Table 2, Fig 1A and 1B). The prevalence of DEC during the winter and spring (the dry season) and the summer (the rainy season) was 38% (124/326) and 39.3% (59/150), respectively. DEC was detected (p = 0.020) in 40.7% (159/391) of diarrheic and 28.2% (24/85) of non-diarrheic children under age five (Table 3). The predominant DEC pathotype was EAEC (41.5%, 76/183), followed by ETEC (21.3%, 39/183%), EPEC (15.3%, 28/183), EIEC (12.6%, 23/183), hybrid strains (7.1%, 13/183), STEC (1.6%, 3/183), and DAEC (0.6%, 1/183). The hybrid pathotypes of DEC isolated in the present study were 5 ETEC/EAEC hybrids, 5 ETEC/EPEC hybrids, and 3 EPEC/EAEC hybrids. Sample gel image for DEC pathotypes is available in S1 Fig.

**Table 2. Demographic factors associated with DEC positive children under age five in Addis Ababa and Debre Berhan, Ethiopia.**

| Variables | | DEC | | Univariate analysis | | Multivariate analysis | |
|---|---|---|---|---|---|---|---|
| | | Yes (n = 183) N (%) | No (n = 293) N (%) | COR (95%CI) | P-value | AOR (95%CI) | P-value |
| Study area | Debre Berhan | 76 (41.5%) | 123 (42.0%) | 1.019 (0.700, 1.481) | 0.923 | 1.075 (0.640, 1.806) | 0.784 |
| | Addis Ababa | 107 (58.5%) | 170 (58.0%) | 1.00 | | 1.00 | |
| Sex | Male | 104 (56.8%) | 170 (58.0%) | 1.050 (0.723, 1.525) | 0.798 | 1.184 (0.782, 1.770) | 0.409 |
| | Female | 79 (43.2%) | 123 (42.0%) | 1.00 | | 1.00 | |
| Age | 0–12 months | 31(16.9%) | 48 (16.4%) | 0.943 (0.566, 1.571) | 0.946 | 0.929 (0.527, 1.638) | 0.967 |
| | 13–24 months | 43 (23.5%) | 66 (22.5%) | 0.935 (0.595, 1.469) | | 1.000 (0.616, 1.622) | |
| | 24–59 months | 109 (59.6%) | 179 (61.1%) | 1.00 | | 1.00 | |
| Seasons | Winter | 84 (45.9%) | 146 (49.8%) | 0.540 (0.351, 0.832) | 0.004 | 0.529 (0.335, 0.835) | 0.009 |
| | Spring | 40 (21.9%) | 56 (19.1%) | 1.084 (0.665, 1.767) | | 0.999(0.597, 1.671) | |
| | Summer | 59 (32.2%) | 91 (31.1%) | 1.00 | | 1.00 | |
| Family income (monthly) | <$36 (low) | 61 (33.3%) | 121 (41.3%) | 2.212 (1.447, 4.118) | 0.002 | 2.364 (1.354, 4.126) | 0.008 |
| | $37-115(middle) | 74 (40.4%) | 133 (45.4% | 2.212 (1.329, 3.681) | | 2.042 (1.191, 3.502) | |
| | > $115(high) | 48 (26.2%) | 39 (13.3%) | 1.00 | | 1.00 | |
| Child care by | House worker | 70 (38.3%) | 142 (48.5%) | 0.742 (0.385, 1.432) | 0.002 | 0.731 (0.366, 1.458) | 0.009 |
| | Mother | 98 (53.6%) | 110 (37.5%) | 0.411 (0.214, 0.787) | | 0.423 (0.211, 0.846) | |
| | Others | 15 (8.2%) | 41 (14.0%) | 1.00 | | 1.00 | |
| Begin supplementary food | <6months | 22 (12.0%) | 59 (20.1%) | 2.458 (0.947, 6.379) | 0.059 | 3.660 (1.322, 10.128) | 0.019 |
| | 6-12months | 150 (82.0%) | 222 (75.8%) | 1.357 (0.583, 3.155) | | 1.826 (0.740, 4.508) | |
| | >12months | 11 (6.0%) | 12 (4.1%) | 1.00 | | 1.00 | |
| Domestic animals | Yes | 69 (37.7% | 156 (53.2%) | 1.881 (1.291, 2.742) | 0.001 | 1.555 (1.016, 2.381) | 0.042 |
| | No | 114 (62.3%) | 137 (46.8%) | 1.00 | | 1.00 | |
| Obtaining water | In shift | 107 (58.5%) | 198 (67.6%) | 1.480 (1.010, 2.170) | 0.044 | 1.735 (1.046, 2.879) | 0.033 |
| | Daily | 76 (41.5%) | 95 (32.4%) | 1.00 | | | |

**COR,** Crude Odd Ration; AOR, Adjusted Odd Ratio; Other, including grandmother, close family member, and day care

Children under age five were more likely to acquire DEC during the summer compared to the winter (AOR = 0.529, CI = 0.335; 0.835) (Table 2). Children who lived in a family with a monthly income of below $36 were more likely to acquire DEC (AOR = 2.364, CI = 1.354; 4.126) compared to children who lived in a family with a monthly income greater than $115. Children who were cared for by their mother had a lower risk of DEC (AOR = 0.423, CI = 0.211; 0.846) compared to those cared for by others, including grandmother, close family member, and day-care. Children under age five who began their supplementary food before six months were more likely to acquire DEC (AOR = 3.660, CI = 1.322; 10.128) than those older than 12 months. Children that lived in a compound with domestic animals had two times the likelihood (AOR = 1.555, 1.016; 2.381) of being positive for DEC compared to those without contact with domestic animals. Access to a clean water supply was significantly associated with acquiring DEC. Children whose families obtained water in shift had more likely to get DEC (AOR = 1.735, OR = 1.046; 2.879) compared to those with a continuous daily supply of water.

## Antimicrobial resistance (AMR) profile

The DEC pathotypes were resistant to ampicillin (95.1%, 174/183) and tetracycline (91.3%, 167/183), ciprofloxacin (14.2%, 26/183), ceftazidime and cefotaxime (16.4%, 30/183), and tri-methoprim-sulfamethoxazole (42.6%, 78/183). ETEC, EIEC, EAEC, and hybrid strains showed

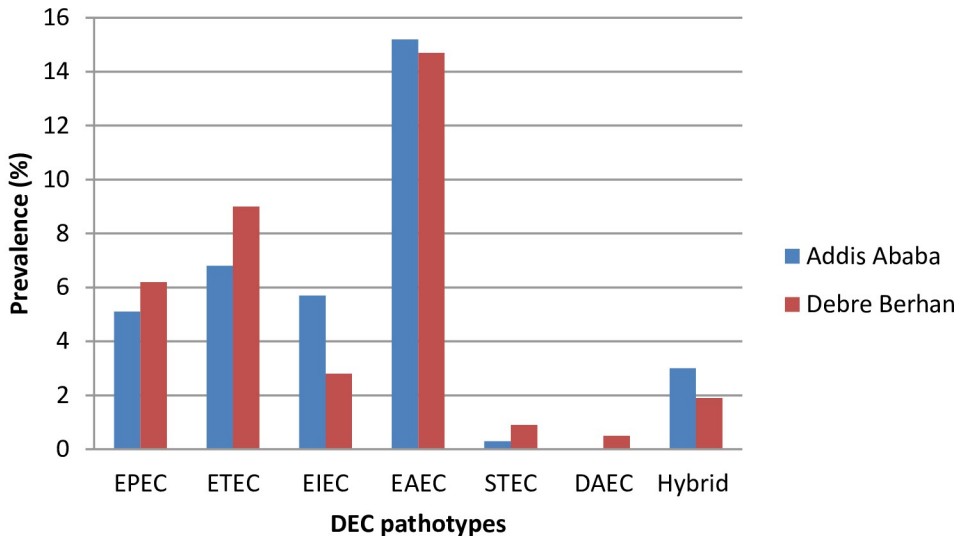

*(a).*

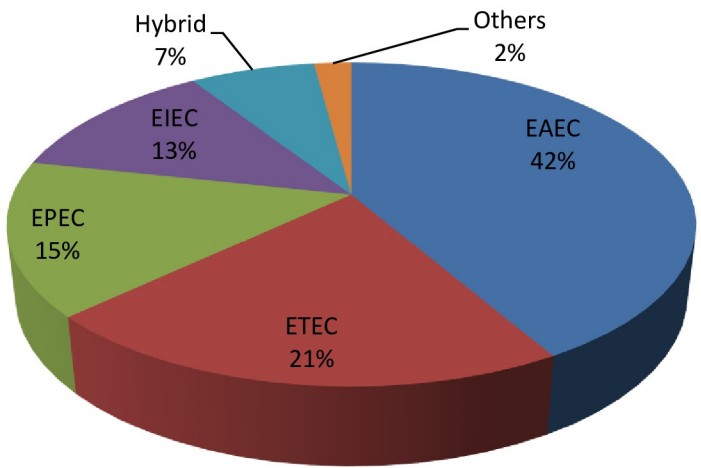

*(b).*

**Fig 1.** **(a)** The distributions of DEC pathotypes and **(b)** Overall prevalence of DEC pathotypes among children under age five in Addis Ababa and Debre Berhan, Ethiopia. DEC, Diarrheagenic *Escherichia coli*; EPEC, Enteropathogenic *Escherichia coli*; ETEC, Enterotoxigenic *Escherichia coli*; EIEC, Enteroinvasive *Escherichia coli*; EAEC, Enteroaggregative *Escherichia coli*; STEC, Shiga-toxin producing *Escherichia coli*; DEAE, Diffusely adherent *Escherichia coli*.

resistance to trimethoprim-sulfamethoxazole from 31% (4/13) to 58% (44/76). DEC strains resistant to ceftazidime and cefotaxime were higher in diarrheic compared to non-diarrheic children (p = 0.010). All three STECs and 1 DAEC, 15% ETEC (6/39), and 22% EAEC (17/76) strains were resistant to ciprofloxacin. EPEC (11%, 3/28), ETEC (15%, 6/39), EAEC (20%, 15/76), and all STEC were resistant to ceftazidime and cefotaxime. Two ETECs (5%, 2/39) and

**Table 3. Distributions of different DEC pathotypes identified from diarrheic and non-diarrheic children under age five in Addis Ababa and Debre Berhan, Ethiopia.**

| DEC Pathotypes | Diarrheic | | | Non-diarrheic | | | Total diarrheic | Total non-diarrheic |
|---|---|---|---|---|---|---|---|---|
| | AA (n = 95) | DB (n = 64) | Total (n = 159) | AA (n = 12) | DB (n = 12) | Total (n = 24) | (n = 391) | (n = 85) |
| EPEC (eae+) | 14.7% | 18.8% | 16.3% | 8.3% | 8.3% | 8.3% | 6.6% | 2.3% |
| ETEC (lt+) | 12.6% | 23.4% | 17% | 25% | 33.3% | 29.2% | 6.9% | 8.2% |
| ETEC (st+) | 3.2% | 0% | 1.9% | 0% | 0% | 0% | 0.8% | 0% |
| ETEC (st+lt+) | 2.1% | 0% | 1.3% | 0% | 0% | 0% | 0.51% | 0% |
| EIEC (virF+ipaH+) | 15.8% | 9.4% | 13.2% | 16.7% | 0% | 8.3% | 5.4% | 2.3% |
| EAEC (aggR+) | 11.6% | 18.8% | 14.5% | 8.3% | 8.3% | 8.3% | 5.9% | 2.3% |
| EAEC (astA+) | 6.3% | 9.4% | 7.5% | 33.3% | 33.3% | 33.3% | 3.1% | 9.4% |
| EAEC (aggR+astA+) | 24.2% | 12.5% | 8.2% | 0% | 0% | 0% | 7.9% | 0% |
| STEC (eae+stx+) | 1.1% | 3.1% | 1.9% | 0% | 0% | 0% | 0.78% | 0% |
| DAEC (daaF+) | 0% | 0% | 0% | 0% | 8.3% | 4.2% | 0% | 1.2% |
| Hybrid | 8.4% | 4.7% | 6.9% | 8.3% | 8.3% | 8.3% | 2.8% | 2.3% |

DEC, Diarrheagenic *Escherichia coli*; EPEC, Enteropathogenic *Escherichia coli*; ETEC, Enterotoxigenic *Escherichia coli*; EIEC, Enteroinvasive *Escherichia coli*; EAEC, Enteroaggregative *Escherichia coli*; STEC, Shiga-toxin producing *Escherichia coli*; DAEC, Diffusely adherent *Escherichia coli*; AA, Addis Ababa; DB, Debre Berhan

two EAECs (3%, 2/76) were resistant to both meropenem and ertapenem. The majority of isolates were more susceptible to amoxicillin-clavulanate, ciprofloxacin, ceftazidime, cefotaxime, and cefepime than to other antimicrobial compounds (Table 4).

Seventy-nine (43.2%, 79/183) of the DEC pathotypes were MDR (Fig 2). Of the MDR strains, 47% (33/70) were from Debre Berhan, and 40.7% (46/113) were from Addis Ababa (P = 0.242). The predominant MDR were EAEC (55.7%, 44/79), followed by ETEC (21.5%, 17/79), EIEC (8.9%, 7/79), EPEC (5.1%, 4/79), STEC (3.8%, 3/79), hybrid (3.8%, 3/79), and DAEC (1.3%, 1/79). A total of 16.4% (30/183) of the DEC pathotypes (38%, 30/79 among MDR) were ESBL producers. Among the ESBL producers, the predominant pathotypes were EAEC (50%,

**Table 4. Antimicrobial resistant profile of DEC pathotypes (n = 183) isolated from diarrheic and non-diarrheic children under age five in Addis Ababa and Debre Berhan, Ethiopia.**

| Antibiotics class | Antibiotics | EPEC (n = 28) | | ETEC (n = 39) | | EIEC (n = 23) | | EAEC (n = 76) | | Hybrid (n = 13) | | Others (n = 4) | |
|---|---|---|---|---|---|---|---|---|---|---|---|---|---|
| | | R | S | R | S | R | S | R | S | R | S | R | S |
| Penicillin | Ampicillin | 89% | 11% | 92% | 8% | 91% | 9% | 99% | 1% | 100% | 0% | 100% | 0% |
| Beta lactam combination | Amoxicillin-Clavulanate | 7% | 93% | 10% | 90% | 4% | 96% | 11% | 90% | 0% | 100% | 0% | 100% |
| Fluoroquinolone | Ciprofloxacin | 4% | 96% | 15% | 85% | 9% | 91% | 22% | 78% | 8% | 92% | 0% | 100% |
| Aminoglycoside | Gentamicin | 7% | 93% | 26% | 74% | 13% | 87% | 41% | 59% | 15% | 85% | 100% | 0% |
| Sulphonamides | trimethoprim-sulfamethoxazole | 14% | 86% | 33% | 67% | 39% | 41% | 58% | 42% | 31% | 69% | 100% | 0% |
| Phenicol | Chloranphenicol | 7% | 93% | 23% | 77% | 9% | 91% | 37% | 63% | 23% | 77% | 100% | 0% |
| Tetracycline | Tetracycline | 89% | 11% | 92% | 8% | 91% | 9% | 92% | 8% | 85% | 15% | 100% | 0% |
| Cephalosporin | Ceftazidime | 11% | 89% | 15% | 85% | 9% | 91% | 20% | 80% | 8% | 92% | 75% | 25% |
| | Cefotaxime | 11% | 89% | 15% | 85% | 9% | 91% | 20% | 80% | 8% | 92% | 75% | 25% |
| | Cefepime | 4% | 96% | 3% | 97% | 0% | 100% | 7% | 93% | 8% | 92% | 0% | 100% |
| Carbapenem | Meropenem | 0% | 100% | 5% | 95% | 0% | 100% | 3% | 97% | 0% | 100% | 0% | 100% |
| | Ertapenem | 0% | 100% | 5% | 95% | 0% | 100% | 3% | 97% | 0% | 100% | 0% | 100% |

DEC, Diarrheagenic *Escherichia coli*; EPEC, Enteropathogenic *Escherichia coli*; ETEC, Enterotoxigenic *Escherichia coli*; EIEC, Enteroinvasive Escherichia coli; EAEC, Enteroaggregative *Escherichia coli*; Others, Shiga-toxin producing *Escherichia coli (*STEC) and Diffusely adherent *Escherichia coli* (DAEC), R, Resistance; S, Susceptible

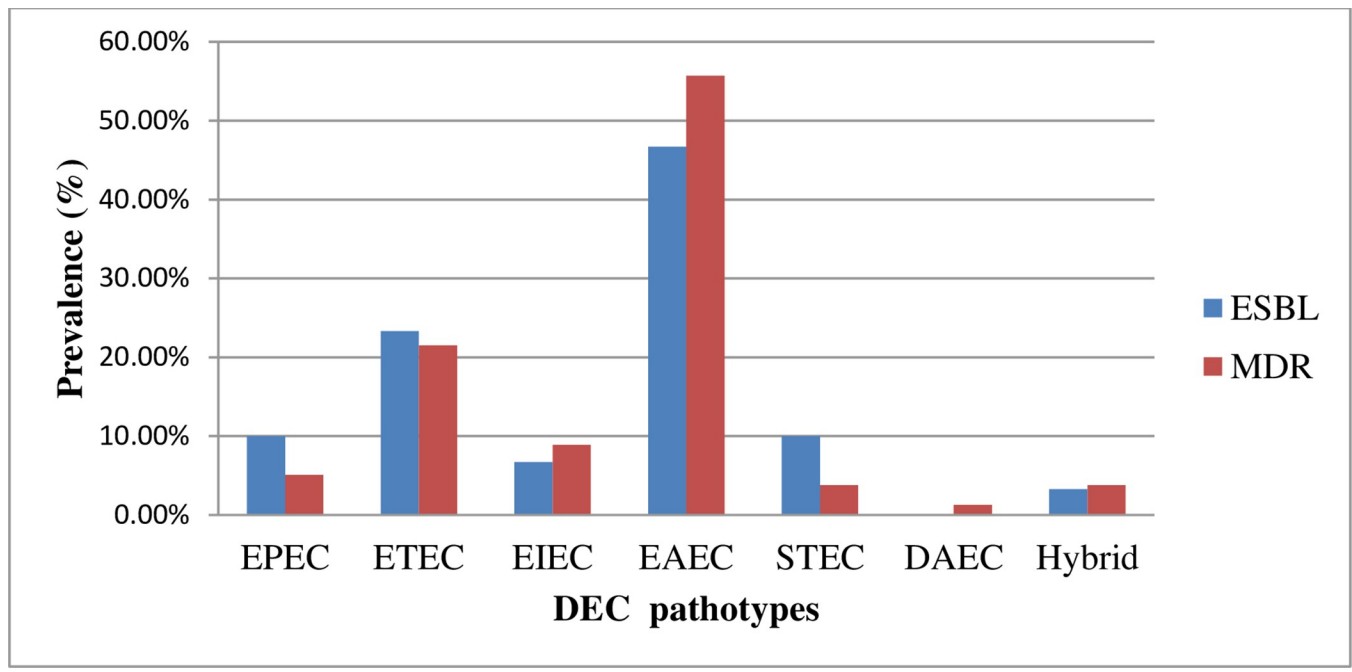

**Fig 2. Extended spectrum beta lactamase (ESBL) producing and multi-drug resistant (MDR) diarrheagenic *E. coli* pathotypes in Addis Ababa and Debre Berhan.** DEC, Diarrheagenic *Escherichia coli*; EPEC, Enteropathogenic *Escherichia coli*; ETEC, Enterotoxigenic *Escherichia coli*; EIEC, Enteroinvasive *Escherichia coli*; EAEC, Enteroaggregative *Escherichia coli*; STEC, Shiga-toxin producing *Escherichia coli*; DEAE, Diffusely adherent *Escherichia coli*.

15/30), followed by ETEC (20%, 6/30), EPEC (10%, 3/30), STEC (10%, 3/30), EIEC (6.7%, 2/30), and the hybrid strains (3.3%, 1/30). A total of 4 (2.2%, 4/183) of the DEC strains were carbapenemase producers, and all were from Addis Ababa.

## Discussion

Diarrhea due to bacterial infection is often self-limiting and does not require identification of the etiological agent for patient management. However, in severe or prolonged cases, with symptoms consistent with invasive disease, or in patients with potential complications, the etiological agents need to be identified for effective treatment [42]. Moreover, public health personnel utilize epidemiological data to identify and track the causes of outbreaks of diarrheal diseases. Such data are scarce in resource-limited countries including Ethiopia, due to either the limitation of the common diagnostic methods to detect DEC [43] or limited applications of currently available diagnostic methods in routine laboratories [44]. In Ethiopia, the molecular epidemiology of DEC and its resistance profile is not well known and the present study determines the occurrence and distribution of DEC pathotypes with their AMR profile, and associated factors in children under age five.

*E. coli* is one of the etiological agents for diarrheal diseases and is especially serious in young children below the age of 5 years. DEC is responsible for up to 30–40% of acute diarrhea episodes in children in developing countries [44]. DEC was isolated from 38.4% of children under age five in the present study, and this is slightly lower compared to other reports, including 48% in Sudan [45], 48.6% in Mozambique [46], and 55% in Iran [47]. However, a higher prevalence of DEC was reported in other African countries, where South Africa presented 82% [48] and Nigeria, 73.8% [49]. Other parts of the world report a lower prevalence of DEC, with China at 7.9% [25], Colombia at 9.8% [39], and India at 17.4% [24]. This discrepancy may be

due to geographical differences associated with climate variability, different study population, and water supply that could influence DEC incidence. The prevalence of DEC in Addis Ababa and Debre Berhan was similar despite the demographic and geographic differences between the cities, demonstrating the clinical importance of DEC in Ethiopia as a whole for the management of childhood diarrhea.

A higher prevalence of DEC (40.7%) was detected in diarrheic compared to non-diarrheic children under age five (28.2%) in the present study (p = 0.020). This finding agrees with reports from Nigeria [49], 76.7% and 54.6%, and Iran [47], 90% and 20%, in diarrheic and non-diarrheic, respectively. The high prevalence of DEC among diarrheic cases compared to the healthy controls in the present study could show the clinical importance of DEC in children under age five in Ethiopia. In the non-diarrheic children, EAEC with EAST-1 and ETEC with LT were the predominant DEC pathotypes in the present study. Most EAEC harbor a virulence plasmid called EAEC-probe (CVD432) that encodes both aggR and EAST-1 [50]. EAEC with EAST-1 was predominantly isolated from probe-negative EAEC strains [50, 51]. However, EAEC with EAST-1 was detected in diarrheic [50] and non-diarrheic children [51]. In the present study, the majority of EAEC in non-diarrheic children were EAEC with EAST-1. EAST-1 may not cause diarrhea alone [52] and is not fully known for its role in diarrhea [6]. ETEC was detected in diarrheic and non-diarrheic children [53] and was seen in the present study. This may indicate the presence of possible transmission from both diarrheic and non-diarrheic children to healthy children. Only LT-positive ETEC was found in the non-diarrheic children in the present study. The ST-positive ETEC is more frequently found in severe infections compared to LT-positive [6, 54].

EAEC (41.5%) was the predominant DEC pathotype identified in the present study and this was similar to the reports from Mozambique by Manhique-Coutinho *et al*, with a higher level of 66.3% [46], and Sudan by Saeed *et al*, with 43% [45]. The genetic heterogeneity of EAEC could contribute to its epidemiology [55]. EAEC causes persistent diarrhea in children in endemic regions [6]. EAEC is divided into typical EAEC which expresses aggR and atypical EAEC which lacks aggR [6]. EAEC can cause persistent diarrhea in children in areas where EAEC is endemic [6]. Persistent or chronic infections by EAEC can damage the intestinal epithelium and cause malnutrition in children in developing countries [6]. EAEC alone or in hybrid strains (EAEC/EHEC) caused outbreaks in different areas of the world [6]. In the present study, both typical and atypical EAEC were found and showed the presence of potential strains in the area for outbreaks.

The second most prevalent DEC pathotype was ETEC (21.3%) which was consistent with reports of 17.3% in Nigeria [49], 14.8% in China [25], 18% in Sudan [45], and 26% in Iran [47]. However, there are reports with lower prevalence [39, 56]. Age category, case type, sample-taking methods, study period, and design may contribute to the discrepancy. The present study only detected atypical EPEC, which lacked bfp (bundle-forming pili) and this is similar to that found in Norway [57] and China [25]. EPEC are classified into typical and atypical EPEC strains based on the presence or absence of the *E. coli* adherence factor plasmid (pEAF) which carries the *bfp* gene [6]. Typical and atypical EPEC are not only differed by genetic and virulence characteristics but also by serotype and reservoirs. Humans are the only reservoir for typical EPEC, whereas atypical EPEC is found in both animals and humans [6]. Atypical EPEC is a highly heterogeneous group, more closely related to STEC, and like STEC, these strains appear to be emerging pathogens that cause outbreaks [6]. The detection of atypical EPEC in the present study should be considered an alarm for health-care personnel for further risk identification to prevent the occurrence of an outbreak.

The newly emerging hybrid pathotypes of DEC were also detected in the present study. These hybrids include ETEC/EAEC, ETEC/EPEC, and EPEC/EAEC. The term hybrid

pathotype refers to new combinations of virulence factors among the classic *E. coli* pathotypes [9]. In 2015, an EPEC expressing the ETEC heat-labile toxin was observed in India [11]. The same combinations in the present study were reported in a study done in South Africa [58]. Of the hybrid strains in the present study were detected, 85% (11/13) from diarrheic, and the remaining (15%) were non-diarrheic. Detection of such a combination of virulence factors in *E. coli* circulating in the community may lead to the occurrence of severe disease due to the hyper-virulent strains in the area. The *E. coli* genomic plasticity that results in the emergence of new hybrid strains could result in severe outbreaks [9]. The detection of hybrid strains should be of public health concern that needs to be carefully monitored through a surveillance program at the health-care facility level.

Socio-demographic factors associated with DEC outbreaks in children under age five were seasons, family income, child-care, time of supplementary food starting, availability of domestic animals, and time for obtaining water. In Table 2, the proportion of DEC in the summer (32.2%) looks less than in the winter (45.9%). However, the actual prevalence of DEC was 36.5% (84/230) in the winter and 39.3% (59/150) in the summer. And it was statistically significant that children under age five were more likely to acquire DEC during the summer (rainy season) compared to winter (dry season) in the present study. The present study shows children visiting the health facility in the summer due to DEC infections. In agreement with this, studies conducted in China and Mexico reported that seasonal distribution revealed that DEC tended to occur during the summer (rainy season) in children [25, 59]. It has been reported that bacterial-caused diarrhea was high during summer and diarrhea due to viral pathogens was high during winter [60]. Low family income was another factor associated with increased DEC infections, most likely due to the lack of hygiene and sanitation facilities. Children who were cared for by their mothers had statistically significant protection (37.5% DEC negative) than those who were cared for by others (grandmothers, close family members, and daycare) (14.0% DEC negative). This may be due to that the mother is more likely careful than other caregivers, potentially contributing to the incidence of DEC in the children. In addition, children whose mothers began supplemental food for their baby before 6 months were more likely to have DEC compared to those whose mothers started it after 12 months. It could be due to exposure to contaminated food through frequent contact during baby feeding. DEC acquisition with the availability of domestic animals in the compound (37.7%) looks greater than with no animals (62.3%) in the present study. However, it was statistically significant that the availability of domestic animals in the compound causes the children to be more likely to get DEC compared to those with no animals. STEC, ETEC, aEPEC, and EIEC are transmitted feco-orally to humans from animal reservoirs [6]. Children who lived in a family who got water in shift were more likely to get DEC compared to those children in families with daily water supply. A previous study in Ethiopia reported that periodically intermittent piped water supplies and point-of-use contamination of household stored water by *E. coli* were associated with acute diarrhea among children under age five [32]. In Ethiopia, most households in both urban (88%) and rural (92%) areas do not treat their water before drinking, and this likely increases the risk of DEC transmission [31].

Most DEC infections are self-limiting and do not require intervention; however, those with severe, persistent, and invasive diseases due to EPEC, EIEC, ETEC, EAEC, and DAEC may require the use of antimicrobials [6]. There is an increasing concern for the rising incidence of MDR DEC today, including ESBL and carbapenemase-producing strains [61]. The Ethiopian guideline for the treatment of diarrheal diseases recommends antibiotics such as ciprofloxacin, sulfamethoxazole+trimethoprim, and ceftriaxone [62]. However, inappropriate antibiotic utilization and management [63] is a problem in Ethiopia. In the present study, DEC pathotypes were resistant to ampicillin (95.1%, 174/183), tetracycline (91.3%, 167/183), trimethoprim-

sulfamethoxazole (42.6%, 78/183), ciprofloxacin (14.2%, 26/183), and ceftazidime and cefotaxime (16.4%, 30/183). It could be problematic in the management of bacterial-caused diarrhea in Ethiopia [62]. DEC pathotypes were also found resistant to these commonly prescribed drugs in other studies, including in Mozambique [46], China [25], and South Africa [48]. However, compared to other reports [25, 48, 49], in the present study, resistance to trimethoprim-sulfamethoxazole and ciprofloxacin was low prevalence. That means, ciprofloxacin and trimethoprim-sulfamethoxazole relatively could be choices of treatment in Ethiopia following antimicrobial susceptibility tests. DEC resistance to ceftazidime and cefotaxime was significantly higher in diarrheic children compared to non-diarrheic children (p = 0.010). The finding of the present study could be explained by the fact that gut inflammation like inflammatory diarrhea can boost horizontal gene transfer between pathogenic and commensal Enterobacteriaceae [64]. However, the controls used in the present study were small compared to the cases and may need further matched case-control studies to conclude that ESBL-producing DEC pathotypes are more prevalent in diarrheic than non-diarrheic children in Ethiopia.

All the STEC were resistant to ceftazidime and cefotaxime in the present study. The prevalence of MDR in the present study was 43.2% which is consistent with studies done in China [25] and India [24]. ESBLs were produced by 16.4% of the DEC pathotypes predominantly by EAEC (50%) in the present study which agrees with studies done in Gabon [65]. The prevalence of carbapenemase-producing strains was 2.2% and found only in ETEC and EAEC. This low prevalence of carbapenemase-producing strains is inconsistent with studies done in China, with 14.3% [25].

## Conclusions

All six common DEC pathotypes that have the potential to cause diarrheal outbreaks were found in Ethiopia, with the predominant being EAEC, followed by ETEC, EPEC, EIEC, hybrid strains, STEC, and DAEC. High resistance to commonly used antibiotics and an increasing number of MDR DEC such as ESBLs and carbapenemase-producing strains will reduce the potential treatment options for DEC infections. In addition, contributing factors for the acquisition of DEC such as low family income and poor childcare were identified. Public health personnel have to understand the presence of virulent and MDR DEC strains among children under age five and take consideration during patient management, and in other prevention and control measures.

## Supporting information

**S1 File. Questionnaire for assessment of factors associated with diarrhea in Addis Ababa and Debre Berhan, Ethiopia (both English and Amharic version).**
(DOCX)

**S1 Fig. Gel images of PCR products of DEC pathotypes.** Lanes: 1 = molecular marker (GeneRuler 100 bp Plus DNA Ladder; Invitrogen Life Technologies); 2 = (negative control, CCUD 24T *Escherichia coli*, 3 and 4 = EIEC, 5 = ETEC, 6 and 7 = EAEC, 8, 9 and 11 = EPEC, and 10 = DAEC. Numbers on the left column denote fragment sizes (bp).
(TIF)

**S1 Table. Clinical presentation of children under age five those were positive for a DEC pathotype during their health-care facility visit.**
(DOCX)

**S1 Raw images.**
(PDF)

## Acknowledgments

We want to thank medical directors and staff in the under-five unit of health-care facilities in Addis Ababa and Debre Berhan for their cooperation during data collection. We also thank the staff working in the Microbiology Laboratory of Tikur Anbessa Specialized Hospital and Debre Berhan, Referral Hospital, and Molecular Biology Laboratory of Armuer Hansen Research Institute for their support during laboratory analysis.

## Author Contributions

**Conceptualization:** Tizazu Zenebe Zelelie, Adane Mihret, Tamrat Abebe.

**Data curation:** Tizazu Zenebe Zelelie, Berhanu Yitayew, Dessalegn Abeje, Ashenafi Alemu, Aminu Seman, Jana Jass, Adane Mihret, Tamrat Abebe.

**Formal analysis:** Tizazu Zenebe Zelelie, Tadesse Eguale, Berhanu Yitayew, Aminu Seman, Jana Jass, Adane Mihret, Tamrat Abebe.

**Investigation:** Tizazu Zenebe Zelelie, Dessalegn Abeje, Ashenafi Alemu, Aminu Seman, Adane Mihret, Tamrat Abebe.

**Methodology:** Tizazu Zenebe Zelelie, Jana Jass, Tamrat Abebe.

**Supervision:** Tadesse Eguale, Tamrat Abebe.

**Validation:** Jana Jass, Adane Mihret, Tamrat Abebe.

**Writing – original draft:** Tizazu Zenebe Zelelie.

**Writing – review & editing:** Tizazu Zenebe Zelelie, Tadesse Eguale, Berhanu Yitayew, Jana Jass, Adane Mihret, Tamrat Abebe.

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
