## [Decision Letter · Decision Letter 0]

21 Nov 2022

PONE-D-22-27001Molecular epidemiology and antimicrobial susceptibility of diarrheagenic Escherichia coli isolated from children under-five with diarrhea and healthy controls in Central EthiopiaPLOS ONE

Dear Dr. Zelelie,

Thank you for submitting your manuscript to PLOS ONE. After careful consideration, we feel that it has merit but does not fully meet PLOS ONE’s publication criteria as it currently stands. Therefore, we invite you to submit a revised version of the manuscript that addresses the points raised during the review process.

We look forward to receiving your revised manuscript.

Kind regards,

Samer Singh, Ph.D.

Academic Editor

PLOS ONE

Journal Requirements:

A clean copy of the edited manuscript (uploaded as the new *manuscript* file).

Additional Editor Comments:

The manuscript has merit. However, the data presentation and representation in the manuscript need to be greatly improved. As reviewers briefly stated, the authors would like to work on the data's logical, precise, and accurate presentation and interpretation to ensure that the message is correctly understood. Additionally, authors may like to get help with grammar and language. 

Reviewers' comments:

Reviewer's Responses to Questions

**Comments to the Author**

1. Is the manuscript technically sound, and do the data support the conclusions?

Reviewer #1: Partly

Reviewer #2: Partly

2. Has the statistical analysis been performed appropriately and rigorously? 

Reviewer #1: Yes

Reviewer #2: Yes

3. Have the authors made all data underlying the findings in their manuscript fully available?

Reviewer #1: Yes

Reviewer #2: Yes

4. Is the manuscript presented in an intelligible fashion and written in standard English?

Reviewer #1: Yes

Reviewer #2: No

5. Review Comments to the Author

Reviewer #1: TITLE:

The title is misleading as it indicates that healthy controls were used; however, the methods imply that children without diarrhea were enrolled, without referring to their health status. Having no diarrhea is not indicative of sound health, as all the children enrolled into the study were seeking care. So, the title needs to be revised to depict exactly what the study was about in context of children’s health.

ABSTRACT:

The abstract indicates;

- That bacterial identification, antibiotic sensitivity testing, and pathotype determination were determined with PCR however, this is not correct especially for bacterial identification and antibiotic sensitivity testing that were phenotypically determined. It’s only DEC pathotypes that were molecularly determined. Revise.

- That 476 specimens were analyzed; were this one specimen per child? How many specimens from each child were analyzed? Revise.

- Results for all the children are grouped together, yet as per the title and/or methods, the study seems to have subgroups i.e., healthy children / children without diarrhea vs. sick children / children with diarrhea. It would be proper to first report findings per subgroup.

- Percentage figures with and without corresponding fractions; it would be better to consistently use percentages with their corresponding fractions.

- Change ‘The overall prevalence rate -------’ to ‘The overall prevalence -------’

INTRODUCTION:

In the second paragraph, revise,

- ‘Diarrheagenic E. coli (DEC) is the major causes of diarrhea -----’ to ‘Diarrheagenic E. coli (DEC) is the major cause of diarrhea -----’

- ‘EPEC is an attaching and effacing (A/E) pathogens -----’ to ‘EPEC is an attaching and effacing (A/E) pathogen -----’

- The acronyms ‘tEPEC’ and ‘aEPEC’ are not defined at their first use

MATERIALS AND METHODS:

- In the section ‘Study area and participants’ nowhere is it mentioned that healthy control children were enrolled as depicted in the title; revise.

- If healthy control children were indeed recruited, the specimen collection procedure described is for only sick children; could you describe the specimen collection procedure for healthy children?

- ESBLs, and so are several acronyms in this manuscript, are introduced without definition at first use.

- As well, ESBLs and carbapenemases are introduced without context and/or rationale, and the procedure for their detection is inadequate.

RESULTS:

- Review presentation for the results of the ESBL- and carbapenemase producing isolates; it would be less confusing if the data for ESBL- and carbapenemase- producing isolates are presented first before showing comparisons with DEC and antibiotic sensitivity data.

- Nothing is mentioned about carbapenemase producing isolates yet the methods are described. Revised.

Reviewer #2: Reviewer’s report

Molecular epidemiology and antimicrobial susceptibility of diarrheagenic Escherichia coli isolated from children under-five with diarrhea and healthy controls in Central Ethiopia by Zenebe et al.

The research article of Zenebe et al. described “Molecular epidemiology and antimicrobial susceptibility of diarrheagenic Escherichia coli isolated from children under-five with diarrhea and healthy controls in Central Ethiopia”. Moderately, the research work was well executed and the manuscript was well written. However, there are some fundamental observations that I have on the way data were presented and interpreted in the manuscript. There are lots of errors, misinterpretations, and misinformation of data in the article. In addition, there are lots of grammatical/typographical errors in the manuscript that require serious overhauling. My comments and observations are as listed below:

Major Comments:-

1. Under the Methodology section, sub-section: “Detection of DEC virulence genes” on page 9: The author gave only one PCR condition for the three different multiplex PCR done. Is the same PCR condition used for the three multiplex reactions? Is the annealing temperature of 60 oC used for the PCR processes all through the experiments? The author needs to clarify this and probably give additional information on the process.

2. Under the Result section, sub-section: “Epidemiology of DEC and associated factors” on page 12:

i. The statement – “Under-five children who visited a health-care facility during winter had a less likelihood…” seems to be ambiguous. Is the author saying that visitation of clinics during the summer periods brings about higher infection with the DEC compared to the winter, as if it is hospital-acquired? Or, is the author trying to say that there is a higher infection rate in the summer than in the winter thereby resulting in many children visiting the clinic due to DEC infections?

ii. The statement – “Children who lived in a family with a monthly income of below $36

were more likely to acquire DEC (AOR=2.364, CI=1.354, 4.126) compared…” seems to be misrepresenting the data presented in Table 2. Family income range of $37-115(middle) actually has a higher number of children positive for DEC (74) compared to 61 recorded for family income of <$36 (low).

iii. The statement – “Children who were cared for by their mother had a lower risk of DEC (AOR=0.423, CI=0.211, 0.846) compared…” Like above, the statement seems not tally with the value presented in Table 2 where a higher number of children taken care of by their mother (98/183, 53.6%) actually came down with DEC infections.

iv. The statement – “Under-five children who began their supplementary food before 6 months old were more likely to acquire DEC (AOR=3.660, CI=1.322, 10.128) than those….” Same as above, the misinterpretation of actual data as expressed in Table 2. Children who began supplementary food at 6-12months have a higher incidence of DEC infections of 150 (82.0%).

v. The statement – “Children that lived in a compound with domestic animals had 2 times the likelihood (AOR=1.555, 1.016, 2.381) of being positive for DEC…” Same as above, misinterpretation of actual data expressed in Table 2. Children who were not having contact with domestic animals showed higher cases of DEC infections, precisely 114 (62.3%) compared to 69 (37.7%) observed for children that lived in a compound with domestic animals.

It could be observed that the author wrote this last statement correctly based on the data expressed in Table 2: “Children whose family obtained water in shift had a greater change of acquiring DEC (AOR=1.735, OR=1.046, 2.879) compared to those with a continuous daily supply of water.” I wondered why all other ones highlighted above were done differently as if the author interpreted the data with premeditated notions.

It could also be noted that the author used these wrongly expressed results to discuss findings from this research work under the Discussion section.

3. Under the Result section, sub-section: “Antimicrobial resistance (AMR) profile” on page 15 and Table 4 on page 16:

It could be observed that data for STEC and DEAE are conspicuously missing in Table 4, whereas, their values were included in the n=183 that was used to arrive at the values presented in the table. Likewise, the correctness of the values presented in the table should be properly checked. For example, why is the author not specific about the number of DEC isolates that were resistant to ampicillin by writing ≥85% on Line 1 under the sub-section?

Also in Table 4 EAEC = 76, whereas, if values for EAEC are summed up in Table 3, it is equal to 86. It seems there are lots of errors in the data presented in the manuscript.

Other Comments:

ABSTRACT SECTION

Under methods, last line, page 2: Consider revising the statement “Bacterial identification, antimicrobial susceptibility, and pathotype determination using polymerase chain reaction (PCR) were evaluated” to “Bacterial isolation, identification, antimicrobial susceptibility, and pathotype determination using polymerase chain reaction (PCR) were done”.

Under results, line 7, page 2: delete “of” before “non-diarrheic”

INTRODUCTION SECTION

Paragraph 2, Last sentence, page 4: The statement: “Perhaps state that recently mixed pathotypes have been identified that have virulence factors associated with more than one pathotype.” It seems something is missing in the statement or it needs to be revised to make better meaning. Also, I think cited reference “(7)” is missing there too.

Paragraph 3, Line 5-7, page 4: Same as above, the statement: “DEC pathotypes cause different outbreaks include in Germany in 2011(15) and Japan in 2016 (16), a haemolytic uremic syndrome caused by STEC, and in Nottingham, UK in 2014 due to EIEC (17)” needs revision.

MATERIALS AND METHODS SECTION

Sub-section: Specimen collection- third to the last line on page 7, add “isolation” after bacterial”

Sub-section: Bacterial Identifications on page 7 should be corrected to Bacterial Isolation and Identifications

Sub-section: Detection of DEC virulence genes, paragraph 2, line 2, page 8: the statement “Briefly, detection of DEC virulence genes were performed…” Should be corrected as: "Briefly, the detection of DEC virulence genes was performed..."

Sub-section: Detection of DEC virulence genes, paragraph 2, line 3, page 8: “…using the primer oligonucleotide DNA sequences…” should be corrected as “…using the primer sequences…”

Sub-section: Detection of DEC virulence genes, paragraph 2, last statement, page 9: the statement: “Known and whole genome sequenced positive DEC strains were used during optimization of the PCR assay.” is not clear.

DISCUSSION SECTION

Paragraph 1, Line 1, page 17: The statement: In Ethiopia, molecular epidemiology of DEC and its resistance profile is unknown” sounds to be outrageous. What about works done by Getaneh et al., 2021; Belete et al., 2022 and so on? Even the author in his systematic review article (Zenebe et al., 2020) gave some insights into the occurrence of DEC in children in Ethiopia. The authors have even failed to give credit to any of the previous works done on E. coli/DEC in Ethiopia, maybe so as to prove the assertion made here. If so, it is grossly unacceptable/unscientific.

Paragraph 5, Line 6, page 19:

The statement: “EPEC classified in to typical…” should be revised as “EPEC are classified into typical...”

Paragraph 7, Line 3-5, page 20: The statement: “In the present study children who visited a

health-care facility during winter were less likely to be positive for DEC (table 2) compared to

children who visited during summer.” Should be revised to make an intended meaning.

Paragraph 7, Line 6, page 20: “Mexco” should be corrected

Paragraph 7, Line 7/8, page 20: The statement “It has been reported that bacterial caused diarrhea….” Could be written as “It has been reported that bacterial-caused diarrhea…” Same thing with “…viral caused diarrhea…” in Line 8, could be written as “…viral-caused diarrhea…”

Paragraph 7, Line 11, page 20: “protective” should be corrected as “protected”

Paragraph 8, Line 12-14, page 21: What is the significance of the statement “DEC pathotypes isolated from diarrheic children showed significantly higher resistance to ceftazidime and cefotaxime compared to those from non-diarrheic children (p=0.010).”? It should be stated.

6. PLOS authors have the option to publish the peer review history of their article (what does this mean?). If published, this will include your full peer review and any attached files.

Reviewer #1: **Yes: **David Patrick Kateete

Reviewer #2: No

---

## [Author Response · Author response to Decision Letter 0]

12 Dec 2022

'Response to Reviewers'

For all reviewers:

We have learnt a lot from your comments and your comments make our manuscript more important than it was. And for this we kindly provide our gratitude to you for your professional supports. Below we go points to points. 

Reviewer #1: TITLE:

Comments: The title is misleading as it indicates that healthy controls were used; however, the methods imply that children without diarrhea were enrolled, without referring to their health status. Having no diarrhea is not indicative of sound health, as all the children enrolled into the study were seeking care. So, the title needs to be revised to depict exactly what the study was about in context of children’s health.

Responses:

• Based on the comment we revised the title. ‘Molecular epidemiology and antimicrobial susceptibility of diarrheagenic Escherichia coli isolated from children under-five with and without diarrhea in Central Ethiopia’ 

• Some DEC pathotypes could be detected from non-diarrheic children and to see this we take non-diarrheic children who came with other medical conditions. And we explained it in the method section. 

Comments: ABSTRACT:

The abstract indicates;

- That bacterial identification, antibiotic sensitivity testing, and pathotype determination were determined with PCR however, this is not correct especially for bacterial identification and antibiotic sensitivity testing that were phenotypically determined. It’s only DEC pathotypes that were molecularly determined. Revise.

Responses:

• Bacterial isolation, identification, antimicrobial susceptibility, and pathotype determination using polymerase chain reaction (PCR) were done.

Comments: - That 476 specimens were analyzed; were this one specimen per child? How many specimens from each child were analyzed? Revise.

Responses:

• One specimen was taken per child and 476 specimens mean total 476 children which were included in the analysis. More explained in method section. 

Comments: Results for all the children are grouped together, yet as per the title and/or methods, the study seems to have subgroups i.e., healthy children / children without diarrhea vs. sick children / children with diarrhea. It would be proper to first report findings per subgroup.

Responses:

• The study has subgroups, children with diarrhea and children without diarrhea. And for this we did separate analysis (table 3). In table 3, DEC pathotypes are presented for case, control, AA, and DB. 

Comments: - Percentage figures with and without corresponding fractions; it would be better to consistently use percentages with their corresponding fractions.

Responses:

• The comment is taken; revised in the manuscript. 

Comments: Change ‘The overall prevalence rate -------’ to ‘The overall prevalence -------’

Responses:

• The overall prevalence… 

Comments: INTRODUCTION:

In the second paragraph, revise,

- ‘Diarrheagenic E. coli (DEC) is the major causes of diarrhea -----’ to ‘Diarrheagenic E. coli (DEC) is the major cause of diarrhea -----’

Responses:

• cause 

Comments: - ‘EPEC is an attaching and effacing (A/E) pathogens -----’ to ‘EPEC is an attaching and effacing (A/E) pathogen -----’

Responses:

• pathogen

Comments: - The acronyms ‘tEPEC’ and ‘aEPEC’ are not defined at their first use

Responses:

• typical EPEC (tEPEC)…… atypical EPEC (aEPEC) (in their first use)

Comments: MATERIALS AND METHODS:

- In the section ‘Study area and participants’ nowhere is it mentioned that healthy control children were enrolled as depicted in the title; revise.

Responses:

• Under-five children with diarrhea who had not received antibiotics were enrolled for the study. Under-five children who were attending the health facility for causes other than diarrhea and without receiving antibiotics treatment during the past three weeks were enrolled as non-diarrheic children.

Comments: - If healthy control children were indeed recruited, the specimen collection procedure described is for only sick children; could you describe the specimen collection procedure for healthy children?

Responses:

• Stool samples were taken with the same procedure from diarrheic and non-diarrheic children. 

Comments: - ESBLs, and so are several acronyms in this manuscript, are introduced without definition at first use.

Responses:

Corrected at their first use

• extended spectrum ß-lactamase (ESBLs)

• polymerase chain reaction (PCR)

• deoxyribonucleic acid

• ATCC (American Type Culture Collection) 

• SPSS (Statistical Package for Social Sciences)

• CCUD (Culture Collection University of Gothenburg)

• EDTA (Ethylenediamine tetraacetic acid) 

Comments: - As well, ESBLs and carbapenemases are introduced without context and/or rationale, and the procedure for their detection is inadequate.

Responses:

• We included a statement in the introduction part: 

o Among MDR strains extended spectrum ß-lactamase (ESBLs) and carbapenemase producing DEC pathotypes are emerging globally (20). Nowadays, there is a serious threat due to ESBL-producing Enterobacteriaceae and an urgent threat due to Carbapenem-resistant E. coli [21]. 

• Both tests were done based on the CLSI guideline (we added the following). 

• Briefly, 1μL loopful colony of test isolate from overnight blood agar plate suspended in 2 mL of nutrient broth (Oxoid UK), and 10μg of meropenem disk immersed to the nutrient broth (fully imersion). The tubes were incubated at 37°C in ambient air without agitation for 4 h ± 15 min. Subsequently, the meropenem disks were removed from the broth using an innoculaton loop. Then, the disks were placed on Mueller-Hinton agar plates (Oxoid, UK) freshly inoculated with a 0.5 McFarland suspension of Escherichia coli ATCC 25922. 

Comments: RESULTS:

- Review presentation for the results of the ESBL- and carbapenemase producing isolates; it would be less confusing if the data for ESBL- and carbapenemase- producing isolates are presented first before showing comparisons with DEC and antibiotic sensitivity data.

Responses:

• Explained in result section 

Comments: - Nothing is mentioned about carbapenemase producing isolates yet the methods are described. Revised.

Responses:

• mentioned in the introduction (see above response)

Reviewer #2: Reviewer’s report

Molecular epidemiology and antimicrobial susceptibility of diarrheagenic Escherichia coli isolated from children under-five with diarrhea and healthy controls in Central Ethiopia by Zenebe et al.

The research article of Zenebe et al. described “Molecular epidemiology and antimicrobial susceptibility of diarrheagenic Escherichia coli isolated from children under-five with diarrhea and healthy controls in Central Ethiopia”. Moderately, the research work was well executed and the manuscript was well written. However, there are some fundamental observations that I have on the way data were presented and interpreted in the manuscript. There are lots of errors, misinterpretations, and misinformation of data in the article. In addition, there are lots of grammatical/typographical errors in the manuscript that require serious overhauling. My comments and observations are as listed below:

Comments: Major Comments:-

1. Under the Methodology section, sub-section: “Detection of DEC virulence genes” on page 9: The author gave only one PCR condition for the three different multiplex PCR done. Is the same PCR condition used for the three multiplex reactions? Is the annealing temperature of 60 oC used for the PCR processes all through the experiments? The author needs to clarify this and probably give additional information on the process.

Responses:

Determination of the DEC pathotypes by PCR was done following a previous procedure with little modification after optimization. During optimization different annealing temperatures (using gradient PCR) were evaluated. For all targets at 60°C annealing temperature was optimal (the same to the previous study). The three PCR reactions were set based on the base pair sizes of the target genes (size difference) according to the manufacturer’s recommendation (amplicol size range). And we set the following PCR reactions, named M-1 to M-3 (mix). 

• M-1: bfp (326bp), eae (917bp) and stx (518bp)

o Base pair difference ≥192bp

• M-2: lt (218bp), st (147bp), virF (618bp) and ipaH (993bp)

o Base pair difference ≥71bp

• M-3: aggR (254bp), astA(111bp) and daaF (542bp).

o ‘Base pair difference ≥143bp

Comments: 2. Under the Result section, sub-section: “Epidemiology of DEC and associated factors” on page 12:

i. The statement – “Under-five children who visited a health-care facility during winter had a less likelihood…” seems to be ambiguous. Is the author saying that visitation of clinics during the summer periods brings about higher infection with the DEC compared to the winter, as if it is hospital-acquired? Or, is the author trying to say that there is a higher infection rate in the summer than in the winter thereby resulting in many children visiting the clinic due to DEC infections?

Responses:

• Under-five children were more likely to acquire DEC during the rainy season compared to dry season (AOR=0.529, CI= 0.335; 0.835) 

Comments: ii. The statement – “Children who lived in a family with a monthly income of below $36

were more likely to acquire DEC (AOR=2.364, CI=1.354, 4.126) compared…” seems to be misrepresenting the data presented in Table 2. Family income range of $37-115(middle) actually has a higher number of children positive for DEC (74) compared to 61 recorded for family income of <$36 (low).

Responses:

• All the results presented in table 2 are based on logistic regression analysis by SPSS. Bivariate and multivariate analyses were done using a binary logistic regression model to identify determinants of DEC acquisition among under-five children. Independent variables for the final model (multivariate logistic regression) were identified using a bivariate logistic regression model with p<0.25. (we included the path used for the analysis, see the revised manuscript)

• Yes, in the actual observation (table2), the value for low family income (<$36) is lower (61) than the value for middle income ($37-115) (74). In the logistic regression analysis done for this study, we use the last variable as reference, in this case high family income ((� $115) and the interpretation was done with this reference. The recorded value may not be necessarily an indicative of statistical value. The actual occurrence of an event (DEC) could be look in terms of odd value. 

• Like for example, calculating the odd value for each (low, middle, and high for DEC positive or negative), it will show the likely occurrence of the event (DEC positive or negative in this case). Odd is the probability of an event occurring divided by the probability of an event not occurring. The higher odd value for an event, is the less likely to occur (vice versa). 

• Based on the above consideration the occurrence of DEC positive in low income is high (low odd value) compare to middle or high income (high odd value). (Odd for DEC positive versus negative). The odd ratio was computed by logistic regression. For more, see responses to v and below.

Comments: iii. The statement – “Children who were cared for by their mother had a lower risk of DEC (AOR=0.423, CI=0.211, 0.846) compared…” Like above, the statement seems not tally with the value presented in Table 2 where a higher number of children taken care of by their mother (98/183, 53.6%) actually came down with DEC infections.

Responses:

• The same explanation made above holds true for this. But in this case, the reference is others (grandmother, close family member, and day care). In compared to others, mother care is protective. 

Comments: iv. The statement – “Under-five children who began their supplementary food before 6 months old were more likely to acquire DEC (AOR=3.660, CI=1.322, 10.128) than those….” Same as above, the misinterpretation of actual data as expressed in Table 2. Children who began supplementary food at 6-12months have a higher incidence of DEC infections of 150 (82.0%).

Responses:

• The same to the above.

Comments: v. The statement – “Children that lived in a compound with domestic animals had 2 times the likelihood (AOR=1.555, 1.016, 2.381) of being positive for DEC…” Same as above, misinterpretation of actual data expressed in Table 2. Children who were not having contact with domestic animals showed higher cases of DEC infections, precisely 114 (62.3%) compared to 69 (37.7%) observed for children that lived in a compound with domestic animals.

Responses:

• The odd for DEC positive with animal is (69/156=0.442); the odd for DEC positive with no animals is (114/137=0.832). The lower the odd is more likely to occur. The odd ratio was computed with logistic regression. 

Comments: It could be observed that the author wrote this last statement correctly based on the data expressed in Table 2: “Children whose family obtained water in shift had a greater change of acquiring DEC (AOR=1.735, OR=1.046, 2.879) compared to those with a continuous daily supply of water.” I wondered why all other ones highlighted above were done differently as if the author interpreted the data with premeditated notions.

Responses: The odd for DEC positive with shift is (107/198=0.540); the odd for DEC positive with daily is (76/95=0.800). ). The lower the odd is more likely to occur. Similar with the above all results. 

It could also be noted that the author used these wrongly expressed results to discuss findings from this research work under the Discussion section.

Comments: 3. Under the Result section, sub-section: “Antimicrobial resistance (AMR) profile” on page 15 and Table 4 on page 16:

It could be observed that data for STEC and DEAE are conspicuously missing in Table 4, whereas, their values were included in the n=183 that was used to arrive at the values presented in the table. Likewise, the correctness of the values presented in the table should be properly checked. For example, why is the author not specific about the number of DEC isolates that were resistant to ampicillin by writing ≥85% on Line 1 under the sub-section?

Also in Table 4 EAEC = 76, whereas, if values for EAEC are summed up in Table 3, it is equal to 86. It seems there are lots of errors in the data presented in the manuscript.

Responses:

• Instead of writing ≥85%, we revised it as ‘ampicillin (95.1%, 174/183) and tetracycline (91.3%, 167/183)

• We revised STEC and DEAE in the table under ‘others’

• In table 3, values for EAEC were incorrect in only column 9 (total calculation error) but correct in other columns. And we corrected it. 

Comments: Other Comments:

ABSTRACT SECTION

Under methods, last line, page 2: Consider revising the statement “Bacterial identification, antimicrobial susceptibility, and pathotype determination using polymerase chain reaction (PCR) were evaluated” to “Bacterial isolation, identification, antimicrobial susceptibility, and pathotype determination using polymerase chain reaction (PCR) were done”.

Under results, line 7, page 2: delete “of” before “non-diarrheic”

Responses: 

• Bacterial isolation, identification, antimicrobial susceptibility, and pathotype determination using polymerase chain reaction (PCR) were done

• in non-diarrheic children

Comments: INTRODUCTION SECTION

Paragraph 2, Last sentence, page 4: The statement: “Perhaps state that recently mixed pathotypes have been identified that have virulence factors associated with more than one pathotype.” It seems something is missing in the statement or it needs to be revised to make better meaning. Also, I think cited reference “(7)” is missing there too.

Responses:

• Recently mixed pathotypes have been identified that have virulence factors associated with more than one pathotypes (7). 

Comments: Paragraph 3, Line 5-7, page 4: Same as above, the statement: “DEC pathotypes cause different outbreaks include in Germany in 2011(15) and Japan in 2016 (16), a haemolytic uremic syndrome caused by STEC, and in Nottingham, UK in 2014 due to EIEC (17)” needs revision.

Response:

• Due to different DEC pathotypes outbreaks were occurred in Germany in 2011(15), in Japan in 2016 (16), and in Nottingham, UK in 2014 (17).

Comments: MATERIALS AND METHODS SECTION

Sub-section: Specimen collection- third to the last line on page 7, add “isolation” after bacterial”

Responses:

• isolation,

Comments: Sub-section: Bacterial Identifications on page 7 should be corrected to Bacterial Isolation and Identifications

Responses:

• bacterial Isolation and Identifications

Comments: Sub-section: Detection of DEC virulence genes, paragraph 2, line 2, page 8: the statement “Briefly, detection of DEC virulence genes were performed…” Should be corrected as: "Briefly, the detection of DEC virulence genes was performed..."

Responses:

• the detection of DEC virulence genes was

Comments: Sub-section: Detection of DEC virulence genes, paragraph 2, line 3, page 8: “…using the primer oligonucleotide DNA sequences…” should be corrected as “…using the primer sequences…”

Responses:

• using the primer sequences

Comments: Sub-section: Detection of DEC virulence genes, paragraph 2, last statement, page 9: the statement: “Known and whole genome sequenced positive DEC strains were used during optimization of the PCR assay.” is not clear.

Responses:

• Previously used positive known DEC strains (whole genome sequenced) were used for optimization of the PCR assay. 

Comments: DISCUSSION SECTION

Paragraph 1, Line 1, page 17: The statement: In Ethiopia, molecular epidemiology of DEC and its resistance profile is unknown” sounds to be outrageous. What about works done by Getaneh et al., 2021; Belete et al., 2022 and so on? Even the author in his systematic review article (Zenebe et al., 2020) gave some insights into the occurrence of DEC in children in Ethiopia. The authors have even failed to give credit to any of the previous works done on E. coli/DEC in Ethiopia, maybe so as to prove the assertion made here. If so, it is grossly unacceptable/unscientific.

Responses:

• We used ‘not well characterized’ in place of unknown. We took the comments. 

Comments: Paragraph 5, Line 6, page 19: The statement: “EPEC classified in to typical…” should be revised as “EPEC are classified into typical...”

Responses:

• EPEC are…

Comments: Paragraph 7, Line 3-5, page 20: The statement: “In the present study children who visited a

health-care facility during winter were less likely to be positive for DEC (table 2) compared to

children who visited during summer.” Should be revised to make an intended meaning.

Responses:

• Under-five children were more likely to acquire DEC during the summer compared to the winter in the present study, resulting in many children visiting the health facility in the summer due to DEC infections. 

Comments: Paragraph 7, Line 6, page 20: “Mexco” should be corrected

Responses:

• Mexico

Comments: Paragraph 7, Line 7/8, page 20: The statement “It has been reported that bacterial caused diarrhea….” Could be written as “It has been reported that bacterial-caused diarrhea…” Same thing with “…viral caused diarrhea…” in Line 8, could be written as “…viral-caused diarrhea…”

Responses:

• It has been reported that bacterial-caused diarrhea was high during summer and viral-caused diarrhea…

Comments: Paragraph 7, Line 11, page 20: “protective” should be corrected as “protected”

Responses:

• protected

Comments: Paragraph 8, Line 12-14, page 21: What is the significance of the statement “DEC pathotypes isolated from diarrheic children showed significantly higher resistance to ceftazidime and cefotaxime compared to those from non-diarrheic children (p=0.010).”? It should be stated.

Responses:

• we revised it. 

• The finding of the present study could be explained by the fact that gut inflammation like inflammatory diarrhea can boost horizontal gene transfer between pathogenic and commensal enterobacteriaceae [55]. However, the comparative groups used in the present study were small compared to cases and it may need further matched case control study for concluding that ESBL producing DEC pathotypes are more prevalent in diarrheic than non-diarrheic children in Ethiopia.

---

## [Decision Letter · Decision Letter 1]

2 Jan 2023

PONE-D-22-27001R1Molecular epidemiology and antimicrobial susceptibility of diarrheagenic Escherichia coli isolated from children under-five with and without diarrhea in Central EthiopiaPLOS ONE

Dear Dr. Zelelie,

Thank you for submitting your manuscript to PLOS ONE. After careful consideration, we feel that it has merit but does not fully meet PLOS ONE’s publication criteria as it currently stands. Therefore, we invite you to submit a revised version of the manuscript that addresses the points raised during the review process.

A thorough revision of the provided methodology, results presentation and interpretation is desired. The points raised by the esteemed reviewer are indicative. Authors would like to check the manuscript thoroughly and amend it accordingly.

We look forward to receiving your revised manuscript.

Kind regards,

Samer Singh, Ph.D.

Academic Editor

PLOS ONE

Journal Requirements:

Additional Editor Comments :

The authors should pay close attention to the provided methodology details, results and interpretation in addition to that indicated by the reviewer. A thoroughly corrected version is desired.

Reviewers' comments:

Reviewer's Responses to Questions

**Comments to the Author**

1. If the authors have adequately addressed your comments raised in a previous round of review and you feel that this manuscript is now acceptable for publication, you may indicate that here to bypass the “Comments to the Author” section, enter your conflict of interest statement in the “Confidential to Editor” section, and submit your "Accept" recommendation.

Reviewer #1: All comments have been addressed

Reviewer #2: (No Response)

2. Is the manuscript technically sound, and do the data support the conclusions?

Reviewer #1: Yes

Reviewer #2: Yes

3. Has the statistical analysis been performed appropriately and rigorously? 

Reviewer #1: Yes

Reviewer #2: Yes

4. Have the authors made all data underlying the findings in their manuscript fully available?

Reviewer #1: Yes

Reviewer #2: Yes

5. Is the manuscript presented in an intelligible fashion and written in standard English?

Reviewer #1: Yes

Reviewer #2: Yes

6. Review Comments to the Author

Reviewer #1: The authors have addressed all the concerns raised in the original submission. I recommend publication of the current version.

Reviewer #2: Reviewer’s report on the Revised Manuscript: PONE-D-22-27001R1

Molecular epidemiology and antimicrobial susceptibility of diarrheagenic Escherichia coli isolated from children under-five with diarrhea and healthy controls in Central Ethiopia by Zenebe et al.

There is no doubt that the quality of the manuscript has increased considerably following the revisions made by the authors. However, I have some minor observations, comments, and inputs.

1. I would like to use one of my initial comments and the authors’ response to point out my observation below:

“Comments: v. The statement – “Children that lived in a compound with domestic animals had 2 times the likelihood (AOR=1.555, 1.016, 2.381) of being positive for DEC…” Same as above, misinterpretation of actual data expressed in Table 2. Children who were not having contact with domestic animals showed higher cases of DEC infections, precisely 114 (62.3%) compared to 69 (37.7%) observed for children that lived in a compound with domestic animals.

Responses:

• The odd for DEC positive with animal is (69/156=0.442); the odd for DEC positive with no animals is (114/137=0.832). The lower the odd is more likely to occur. The odd ratio was computed with logistic regression.”

My observation/question is, what is the relevance of Table 3 if the data presented in it does not give direct support to the information expressed in the manuscript except an additional calculation is done by the reader of the manuscript in order to understand claims by the authors? For example, in the above comment and response, the authors have to manually calculate another set of odd values (69/156=0.442 and 114/137=0.832) to explain their point that “Children that lived in a compound with domestic animals had 2 times the likelihood (AOR=1.555, 1.016, 2.381) of being positive for DEC…” I think the authors should be able to use the data expressed in the table to write their results and discuss them in a manner that ordinary readers could understand without having to pick calculator to start doing additional calculations.

2. Under the Introduction Section:

i. On Page 4, the revised last statement in paragraph 2, “Recently mixed pathotypes have been identified that have virulence factors associated with more than one pathotype (7).” I am surprised the citation (7) is dated 2012 whereas the authors made the statement “Recently…|”

ii. On Page 5, paragraph 4, line 4, I guessed the word “systemic” is supposed to be “systematic”. The authors should check it up. Also, the word: “presences” should be corrected to “presence” in the last statement of the same paragraph

3. Under the Methodology Section:

i. Sub-section: Phenotypic detection of ESBLs and Carbapenemases production, line 7, page 8. The statement “Briefly, 1μL loopful colony of test isolate from overnight blood agar plate suspended…” should be revised. I can’t figure out how a 1μL loopful colony of test isolate could be taken from an agar medium.

ii. Sub-section: Detection of DEC virulence genes, second to the last statement on page 10. The statement: “Previously used positive known DEC strains (whole genome sequenced) were used for the optimization of the PCR assay.” should be revised for better meaning. Previously used positive strains by who?

7. PLOS authors have the option to publish the peer review history of their article (what does this mean?). If published, this will include your full peer review and any attached files.

Reviewer #1: **Yes: **David Patrick Kateete

Reviewer #2: **Yes: **Kayode Olayinka Afolabi

---

## [Author Response · Author response to Decision Letter 1]

8 Jan 2023

For all reviewers:

We have learnt a lot from your comments and your comments make our manuscript more important than it was. Your additional comment further improves the manuscript. And for this we kindly provide our gratitude to you for your professional supports. Below we go points to points. 

Reviewer’s report on the Revised Manuscript: PONE-D-22-27001R1

Molecular epidemiology and antimicrobial susceptibility of diarrheagenic Escherichia coli isolated from children under-five with diarrhea and healthy controls in Central Ethiopia by Zenebe et al. 

Comments:

There is no doubt that the quality of the manuscript has increased considerably following the revisions made by the authors. However, I have some minor observations, comments, and inputs.

1. I would like to use one of my initial comments and the authors’ response to point out my observation below:

“Comments: v. The statement – “Children that lived in a compound with domestic animals had 2 times the likelihood (AOR=1.555, 1.016, 2.381) of being positive for DEC…” Same as above, misinterpretation of actual data expressed in Table 2. Children who were not having contact with domestic animals showed higher cases of DEC infections, precisely 114 (62.3%) compared to 69 (37.7%) observed for children that lived in a compound with domestic animals.

 Responses:

 • The odd for DEC positive with animal is (69/156=0.442); the odd for DEC positive with no animals is (114/137=0.832). The lower the odd is more likely to occur. The odd ratio was computed with logistic regression.”

My observation/question is, what is the relevance of Table 3 if the data presented in it does not give direct support to the information expressed in the manuscript except an additional calculation is done by the reader of the manuscript in order to understand claims by the authors? For example, in the above comment and response, the authors have to manually calculate another set of odd values (69/156=0.442 and 114/137=0.832) to explain their point that “Children that lived in a compound with domestic animals had 2 times the likelihood (AOR=1.555, 1.016, 2.381) of being positive for DEC…” I think the authors should be able to use the data expressed in the table to write their results and discuss them in a manner that ordinary readers could understand without having to pick calculator to start doing additional calculations.

Responses:

1. Based on the comments given by the reviewer we revised it. We used the data (the season, child care, and availability of animal) expressed in the table in the discussion and make more understandable. 

Comments:

2. Under the Introduction Section:

i. On Page 4, the revised last statement in paragraph 2, “Recently mixed pathotypes have been identified that have virulence factors associated with more than one pathotype (7).” I am surprised the citation (7) is dated 2012 whereas the authors made the statement “Recently…|”

ii. On Page 5, paragraph 4, line 4, I guessed the word “systemic” is supposed to be “systematic”. The authors should check it up. Also, the word: “presences” should be corrected to “presence” in the last statement of the same paragraph 

Responses:

i. The reference (7), and [49] were misplaced and we corrected them. In place of reference (7) it was reference [8] which is recent data (of 2020). 

ii. We corrected the systemic to systematic; presences to presence. 

Comments:

3. Under the Methodology Section:

i. Sub-section: Phenotypic detection of ESBLs and Carbapenemases production, line 7, page 8. The statement “Briefly, 1μL loopful colony of test isolate from overnight blood agar plate suspended…” should be revised. I can’t figure out how a 1μL loopful colony of test isolate could be taken from an agar medium.

ii. Sub-section: Detection of DEC virulence genes, second to the last statement on page 10. The statement: “Previously used positive known DEC strains (whole genome sequenced) were used for the optimization of the PCR assay.” should be revised for better meaning. Previously used positive strains by who?

Responses:

i. We rewrite the statement using the phrase ‘a loopful (by using a 1μL size loop) of the colony of the test isolates’

ii. Based on the comment we revised the statement (highlighted in the manuscript). 

Comments:

 Additional Editor Comments :

The authors should pay close attention to the provided methodology details, results and interpretation in addition to that indicated by the reviewer. A thoroughly corrected version is desired.

Responses:

• The reference (7), and [49] were misplaced and we corrected them. We reviewed the reference list and corrected accordingly. We included DOI number for these papers that have DOI number. 

• We reviewed thoroughly the paper and made all the corrections.

---

## [Decision Letter · Decision Letter 2]

28 Mar 2023

PONE-D-22-27001R2Molecular epidemiology and antimicrobial susceptibility of diarrheagenic Escherichia coli isolated from children under-five with and without diarrhea in Central EthiopiaPLOS ONE

Dear Dr. Zelelie,

Thank you for submitting your manuscript to PLOS ONE. After careful consideration, we feel that it has merit but does not fully meet PLOS ONE’s publication criteria as it currently stands. Therefore, we invite you to submit a revised version of the manuscript that addresses the points raised during the review process.

We look forward to receiving your revised manuscript.

Kind regards,

Samer Singh, Ph.D.

Academic Editor

PLOS ONE

**Additional Editor Comments:**

The manuscript is improved from the last version but still rife with deficiencies, as indicated by a reviewer. The decision of major revision is just to indicate gravity of the minor issues that still needs to be fixed. The points raised by the learned reviewer are indicative; a thorough revision of the manuscript is desired. The authors would like to pay attention to the precise presentation of facts and methodology. Catalog no. of chemicals and required details/ permissions as expected for the study be indicated or provided. The inclusion of relevant details with clarifying statements would tremendously improve its quality.

For language improvement, which could be partially contributing to the indicated issues, the help of native speakers may be sought by the authors.

Reviewers' comments:

Reviewer's Responses to Questions

**Comments to the Author**

1. If the authors have adequately addressed your comments raised in a previous round of review and you feel that this manuscript is now acceptable for publication, you may indicate that here to bypass the “Comments to the Author” section, enter your conflict of interest statement in the “Confidential to Editor” section, and submit your "Accept" recommendation.

Reviewer #2: All comments have been addressed

Reviewer #3: (No Response)

2. Is the manuscript technically sound, and do the data support the conclusions?

Reviewer #2: (No Response)

Reviewer #3: Partly

3. Has the statistical analysis been performed appropriately and rigorously? 

Reviewer #2: (No Response)

Reviewer #3: Yes

4. Have the authors made all data underlying the findings in their manuscript fully available?

Reviewer #2: (No Response)

Reviewer #3: Yes

5. Is the manuscript presented in an intelligible fashion and written in standard English?

Reviewer #2: (No Response)

Reviewer #3: No

6. Review Comments to the Author

Reviewer #2: (No Response)

Reviewer #3: Major Comments:

The present study was carried out in Ethiopia and showed the prevalence of DEC pathotypes in children under the age of five years. The authors were trying to find out and reported the prevalence of different strains of DEC pathotypes such as EAEC, ETEC, EIEC, and some hybrid strains also. The authors reported the prevalence of MDR bacteria and ESBL pathogens.

1. Diarrheagenic Escherichia coli (DEC) Pl. italicized the Escherichia coli.

2. Diarrheagenic Escherichia coli (DEC) is the major cause of diarrhea in under-five children. Pl. revised the sentences carefully.

3. The present study aims to determine molecular epidemiology and antimicrobial resistance profiles of DEC and identify contributing factors for acquisition among under-five children in Central Ethiopia. What is the mean of under-five children? An author should write it like children under the age of five.

4. Ethiopia is one of the 15 pneumonia and diarrhea are high-burden countries. The author should Pl. revised the sentence.

5. A systemic review and meta-analysis study done in Ethiopia reported a 22% pooled prevalence of diarrhea among under-five children. What do you mean? The author should kindly revise the sentence.

6. A systemic review and meta-analysis study done in Ethiopia reported a 22% pooled prevalence of diarrhea among under-five children. What are under five children? The author should rephrase it.

7. Specific combinations of virulence traits group DEC into six common pathotypes, including enterotoxigenic E. coli (ETEC), enteropathogenic E. coli (EPEC), Shiga-toxin producing E. coli (STEC)/enterohemorrhagic E. coli (EHEC), enteroinvasive E. coli (EIEC), enteroaggregative E. coli (EAEC), and diffusely adherent E. coli (DAEC). The following sentence is not clear. Pl. revised the sentence.

8. EPEC is an attaching and effacing (A/E) pathogen known by the A/E lesion formation. Pl. revised the sentence.

9. Based on the E. coli adherence factor (EAF) plasmid (pEAF) encodes bundle-forming pili (BFP). Pl. revise the sentence properly.

10. EPEC are classified into typical EPEC (tEPEC) (presence of BFP) and atypical EPEC (aEPEC). What do you mean? Pl. revised the manuscript.

11. STEC causes mild to bloody diarrhea, foodborne outbreaks, and severe illness, including hemorrhagic colitis and hemolytic uremic syndrome, and is known by its potent toxin production, Shiga toxin 1 or 2 genes (stx1, stx2). The author should rephrase the sentence and make it clear and comprehensible.

12. Transcription activator (virF). What is the role of virF?

13. Hybrid DEC pathotype strains are increasingly reported elsewhere in the world. What does it mean? Is it elsewhere???

14. These include EPEC/ETEC in India [10], STEC/EAEC in Germany [11] and Indonesia [12], EPEC/STEC [13] and EPEC/EAEC [14] in Brazil, and STEC/ETEC in Sweden [15]. What does it mean? Is EPEC strain also contain ETEC virulence factor?

15. Outbreaks of diarrheal diseases due to different DEC pathotypes had been reported in different countries including Germany in 2011[16], Japan in 2016 [17], and Nottingham, UK in 2014 [18]. Are any recent outbreaks of diarrhea due to DEC pathotypes like in 2018-2022?

16. Among MDR strains extended spectrum ß-lactamase (ESBLs) and carbapenemase-producing DEC pathotypes are emerging globally (20). Typological error. Pl. correct it.

17. The occurrence of hybrid DEC [10-13], outbreak-causing DEC [16-18], and resistant DEC [20, 21] reported so far showed the possible emergence of hyper-virulent DEC strains anywhere to cause severe diseases or outbreaks. The author should Pl. revised the sentence and make it easy to read and understand.

18. Epidemiological data of DEC will assist public health personnel in identifying and tracking bacterial outbreaks and severe diarrheal diseases. Thus, understanding the distribution pattern of enteric pathogens within the community will help control or prevent fatal outbreaks. Authors suddenly jumped from different strains of DAE to epidemiological factors.

19. The author should submit the structured questionnaire as the supplementary file which was used for the study.

20. How Author has prevented the contamination of urine during stool sample collection?

21. Which type of sample container was used for the stool sample collection? Pl. describe.

22. The author should briefly describe how aseptic conditions were maintained during stool sample collection.

23. A large number of typological errors were present throughout the manuscript.

24. How stool samples were inoculated on MacConkey agar?

25. Stool samples were diluted or directly streaked on MacConkey agar?

26. The author should briefly describe the inoculation of stool samples on MacConkey agar.

27. 16% glycerol stock was used?

28. Why author should store the E. coli isolates in Brain Heart Infusion agar? Generally, this agar is used for the cultivation of streptococci, meningococci, etc.

29. Ampicillin concentration was 10mcg?

30. Imipenem concentration was 30 mcg?

31. MDR was defined when the isolate was nonsusceptible to at least one agent among three or more antimicrobial categories. Is this the definition of MDR bacteria? The author should refer to the following paper for reference.

a. Singh AK, Das S, Kumar S, Gajamer VR, Najar IN, Lepcha YD, Tiwari HK, Singh S. Distribution of antibiotic-resistant Enterobacteriaceae pathogens in potable spring water of eastern Indian Himalayas: emphasis on virulence gene and antibiotic resistance genes in Escherichia coli. Frontiers in Microbiology. 2020 Nov 5;11:581072.

b. Singh AK, Das S, Singh S, Gajamer VR, Pradhan N, Lepcha YD, Tiwari HK. Prevalence of antibiotic resistance in commensal Escherichia coli among the children in rural hill communities of Northeast India. PloS one. 2018 Jun 18;13(6):e0199179.

32. What is the positive and negative control used for the Antibiotic Susceptibility test/

33. What are the inclusion and exclusion criteria used for the present study?

34. What are the basic criteria used for the selection of antibiotics?

35. by using a 1μL size loop? What is the standard size of the inoculation loop? Is it 1 μL?

36. A confirmatory test for ESBLs activity was conducted for isolates resistant to cefotaxime (30μg) and ceftazidime (30μg) by the combination disk method. The author should briefly describe the screening test method and antibiotics used in the confirmatory test.

37. E. coli resistance to ceftazidime and cefotaxime were considered ESBL producers.

38. What is the role of clavulanic acid during ESBL detection?

39. Without using clavulanic acid author confirmed the ESBL pathogens.

40. Subsequently, the meropenem disks were removed from the broth using an inoculaton loop. Inoculaton????

41. DEC strains positive for the relevant virulence gene (s) stored in the Armauer Hansen Research Institute from previous works and CCUD (Culture Collection University of Gothenburg) 24T E. coli were used as the positive and negative control, respectively. What is 24T E. coli? The author should describe what is the positive and negative control.

42. An author should briefly and separately describe the number of samples collected both diarrheal as well as non-diarrheal during the dry season and rainy season.

43. All 428 E. coli isolates were tested by PCR, of which 183 (42.8%, 183/428) were positive for DEC pathotype. The overall prevalence of DEC in the present study was 38.4% (183/476), and 58.5% (107/183) of the DEC types were isolated from Addis Ababa and the remaining from Debre Berhan. The line shows contradictory results to each other. The overall prevalence of DEC in the present study was 42.8% but again in the second line author mentioned that the overall prevalence of DEC type is 38.4%. The author should briefly revise the sentence.

44. The overall prevalence of DEC in the present study was 38.4% (183/476), and 58.5% (107/183) of the DEC types were isolated from Addis Ababa and the remaining from Debre Berhan (Table 3, Fig1 a & b). The current result and table 3 data were not clear.

45. In Table 3 what are the total cases meaning? And what is total control?

46. Dry season? What is the meaning of the dry season? The author did not mention any data on the dry season in table 2.

47. Under-five children were more likely to acquire DEC during the rainy season compared to the dry season (AOR=0.529, CI= 0.335; 0.835) (Table 2). In table 2 author mentioned the data of summer, Winter, and Spring whereas in the manuscript it was written as dry season and rainy season as well as some places' rainy season. The author should write it properly in the either rainy and dry seasons and spring, summer, and winter in tabl2 as well as the manuscript write-up.

48. The data of table 2 shows that the most prevalent season for DEC was winter (45.9%) followed by summer (32.2%) and spring (21.9%). Kindly check it.

49. Resistance to ceftazidime and cefotaxime was higher in DEC isolates obtained from diarrheic compared to non-diarrheic children (p=0.010). All the STEC and DAEC, 15% ETEC (6/39), and 22% EAEC (17/76) were resistant to ciprofloxacin. Kindly rephrase the sentence properly.

50. ETEC (5%, 2/39) and EAEC (3%, 2/76) were resistant to meropenem and ertapenem. The author should separately describe what percentage of ETEC as well as EAEC were resistant to meropenem as well as entrapenem.

51. This finding agrees with reports from Nigeria [42], and Iran [40]. The author should add the percentage DEC in diarrheic and non-diarrheic in Nigeria and Iran.

52. Only LT-positive ETEC was found in the non-diarrheic children in the present study. ST-positive ETEC strains are commonly associated with diarrhea compared to LT-positive. LT toxin is not responsible for diarrhea?

53. EAEC (41.5%) was the predominant DEC pathotype identified in the present study. What could be the plausible region?

54. The second most prevalent DEC pathotype was ETEC (21.3%) which was consistent with studies done in Nigeria [42], China [21], Sudan [38], and Iran [40]. The author should kindly add the data of these countries.

55. In table 2, the acquisition of DEC in the rainy season (32.2%) looks less than in the dry season (45.9%). It is quite confusing some tables show data for winter, spring, and summer whereas the author discusses the dry season and rainy season. The author should kindly revise it thoroughly and use either winter, summer as well as spring or dry and rainy seasons. What could be the possible reason for the prevalence of DEC pathotypes in winter as compared to the rainy season? The author should discuss it with suitable reason.

56. In table 2, the acquisition of DEC in the rainy season (32.2%) looks less than in the dry season (45.9%). However, it was statistically significant that under-five children were more likely to acquire DEC during the rainy season compared to the dry season in the present study, resulting in children visiting the health facility in the summer due to DEC infections. If children have visited the health facility center in the rainy season then why is the prevalence of DEC pathotypes shown higher in the winter season?

57. Children who were cared for by their mothers (37.5%) had statistically significant protection than those who were cared for by others (grandmothers, close family members, and daycare) (14.0%). The author should recheck the data properly. It is 37.5% or 53.6%??

58. This may be due to that the mother is more likely careful than other caregivers, potentially contributing to the incidence of DEC in the children. What could be the possible reason for the prevalence of DEC pathotypes in more who cares by mother as compared to other caregivers?

59. Any significant effect of the number of children in a particular family on the prevalence of DEC pathotype in that particular family?

60. What is the meaning of total control which showed in table 3?

61. All 428 E. coli isolates were tested by PCR, of which 183 (42.8%, 183/428) were positive for DEC pathotype. The overall prevalence of DEC in the present study was 38.4% (183/476), and 58.5% (107/183) of the DEC types were isolated from Addis Ababa and the remaining from Debre Berhan (Table 3, Fig1 a & b). The author should kindly revise it properly. It creates confusion like 42.8%% were DEC type and the next line overall prevalence was 38.4%.

62. In table 3 the DAEC prevalence was showing 14.5% whereas in the manuscript it was written as 41.5%. The author should revise the tables and write either in percentages or numbers.

63. The author should revise the manuscript thoroughly and correct the data in the table as well as the manuscript.

7. PLOS authors have the option to publish the peer review history of their article (what does this mean?). If published, this will include your full peer review and any attached files.

Reviewer #2: **Yes: **Kayode Olayinka Afolabi

Reviewer #3: **Yes: **Dr. Ashish Kumar Singh

---

## [Author Response · Author response to Decision Letter 2]

14 Apr 2023

For all academic editor and reviewer(s):

First, we thank you for the comments. We have learnt a lot from your comments and your comments make our manuscript more important than it was. In addition, during the review process we got opportunity to improve the manuscript. Your additional comment further improves the manuscript. And for this we kindly provide our gratitude to you for your professional supports. We thank you to each comments and concerns. Below we go points to points. For some similar comments, we gave responses together by indicating the comment numbers. 

Major Comments:

The present study was carried out in Ethiopia and showed the prevalence of DEC pathotypes in children under the age of five years. The authors were trying to find out and reported the prevalence of different strains of DEC pathotypes such as EAEC, ETEC, EIEC, and some hybrid strains also. The authors reported the prevalence of MDR bacteria and ESBL pathogens. 

1. Diarrheagenic Escherichia coli (DEC) Pl. italicized the Escherichia coli.

• Thank you for the comment. We took the comment and iltalized throughout the manuscript. 

2. Diarrheagenic Escherichia coli (DEC) is the major cause of diarrhea in under-five children. Pl. revised the sentences carefully.

• Based on the comment, we revised the sentence: Among the bacterial agents, diarrheagenic Escherichia coli (DEC) is the major causal agent of diarrhea in children under age five.

3. The present study aims to determine molecular epidemiology and antimicrobial resistance profiles of DEC and identify contributing factors for acquisition among under-five children in Central Ethiopia. What is the mean of under-five children? An author should write it like children under the age of five.

• Again we thank you for the comments. In most literature (scholars) use the term ‘unde-five children’, ‘children under the age of five (the one you recommended)’ , or ‘children under age five’. After reviewing and considering your comment we replaced the ‘under-five children’ by ‘children under age five’ throught out the manuscript. In our case it refere to children with age of below five. 

4. Ethiopia is one of the 15 pneumonia and diarrhea are high-burden countries. The author should Pl. revised the sentence.

• We revised it: Ethiopia ranks fifth among the 15 pneumonia and diarrhea high-burden countries in deaths (due to pneumonia and diarrhea) of children under age five.

5. A systemic review and meta-analysis study done in Ethiopia reported a 22% pooled prevalence of diarrhea among under-five children. What do you mean? The author should kindly revise the sentence.

• We took the comment and made revision: A systematic review and meta-analysis conducted in Ethiopia reported a 22% prevalence of diarrhea among children under age five. 

6. A systemic review and meta-analysis study done in Ethiopia reported a 22% pooled prevalence of diarrhea among under-five children. What are under five children? The author should rephrase it.

• We used the phrase’ children under age five’ instead (see response to comment 3).

7. Specific combinations of virulence traits group DEC into six common pathotypes, including enterotoxigenic E. coli (ETEC), enteropathogenic E. coli (EPEC), Shiga-toxin producing E. coli (STEC)/enterohemorrhagic E. coli (EHEC), enteroinvasive E. coli (EIEC), enteroaggregative E. coli (EAEC), and diffusely adherent E. coli (DAEC). The following sentence is not clear. Pl. revised the sentence.

8. EPEC is an attaching and effacing (A/E) pathogen known by the A/E lesion formation. Pl. revised the sentence.

9. Based on the E. coli adherence factor (EAF) plasmid (pEAF) encodes bundle-forming pili (BFP). Pl. revise the sentence properly.

10. EPEC are classified into typical EPEC (tEPEC) (presence of BFP) and atypical EPEC (aEPEC). What do you mean? Pl. revised the manuscript.

• We revised it by taking comment 8, 9 & 10 above: Based on the presence or absence of E. coli adherence factor plasmid (pEAF), EPEC strains are classified into typical EPEC (tEPEC) that has the pEAF and atypical EPEC (aEPEC) that lacks the pEAF [6]. The plasmid (pEAF) carried a gene called bfp that encodes bundle-forming pili (BFP), an important virulence factor of the tEPEC strains.

11. STEC causes mild to bloody diarrhea, foodborne outbreaks, and severe illness, including hemorrhagic colitis and hemolytic uremic syndrome, and is known by its potent toxin production, Shiga toxin 1 or 2 genes (stx1, stx2). The author should rephrase the sentence and make it clear and comprehensible. 

• STEC is a foodborne and zoonotic pathogen, and it causes non-bloody diarrhea, bloody diarrhea, hemorrhagic colitis, and haemolytic uremic syndrome. STEC is known for its potent toxin called Shiga toxin encoded by stx genes (stx1 and stx2).

12. Transcription activator (virF). What is the role of virF?

• Thank you and we reconsidered it. We included this: The virF is a DNA-binding protein (also called master regulator) that control the expression of virulence factors (e.g. T3SS and effector proteins) by regulating virB (another transcriptional regulator) and icsA genes. VirB and IcsA (VirG) are required to the full expression of the invasion program by EIEC.

13. Hybrid DEC pathotype strains are increasingly reported elsewhere in the world. What does it mean? Is it elsewhere???

• We made a slight modification on the sentence: Hybrid DEC pathotype strains are increasingly reported in different countries of the world.

14. These include EPEC/ETEC in India [10], STEC/EAEC in Germany [11] and Indonesia [12], EPEC/STEC [13] and EPEC/EAEC [14] in Brazil, and STEC/ETEC in Sweden [15]. What does it mean? Is EPEC strain also contain ETEC virulence factor?

• Sorry, if we miss understood your comment? The statement is the continuation (linked) of the prevous sentence and this was to examplify reports of some countries. Hybrid strains are DEC strains that have virulence factors associated with more than one pathotype (explained in the previous paragraph). We revised the sentence: Among the hybrid strains reported include EPEC/ETEC in India [11], STEC/EAEC in Germany [12] and Indonesia [13], EPEC/STEC [14] and EPEC/EAEC [15] in Brazil, and STEC/ETEC in Sweden [16].

15. Outbreaks of diarrheal diseases due to different DEC pathotypes had been reported in different countries including Germany in 2011[16], Japan in 2016 [17], and Nottingham, UK in 2014 [18]. Are any recent outbreaks of diarrhea due to DEC pathotypes like in 2018-2022?

• Based on your comment, we added some recent outbreak reports: South Korea in 2018, Japan in 2020, and United Kingdom in 2020. And we also added this: In addition, outbreaks due to different E. coli strains that occurred annually are available in the Centers for Disease Control and Prevention outbreak reports (https://www.cdc.gov/ecoli/2022-outbreaks.html).

16. Among MDR strains extended spectrum ß-lactamase (ESBLs) and carbapenemase-producing DEC pathotypes are emerging globally (20). Typological error. Pl. correct it.

• We corrected it. From MDR bacterial strains, the extended-spectrum ß-lactamase (ESBL) and carbapenemase-producing DEC pathotypes are among the emerging pathogens globally..

17. The occurrence of hybrid DEC [10-13], outbreak-causing DEC [16-18], and resistant DEC [20, 21] reported so far showed the possible emergence of hyper-virulent DEC strains anywhere to cause severe diseases or outbreaks. The author should Pl. revised the sentence and make it easy to read and understand. 

• Based on the comment we made the revision. The occurrence of hybrid, outbreak-causing, and resistant DEC strains in different areas of the world could reveal the presence of a threat for the possible occurrence of DEC-caused severe diseases or outbreaks.

18. Epidemiological data of DEC will assist public health personnel in identifying and tracking bacterial outbreaks and severe diarrheal diseases. Thus, understanding the distribution pattern of enteric pathogens within the community will help control or prevent fatal outbreaks. Authors suddenly jumped from different strains of DAE to epidemiological factors. 

• This was to mean that providing such data will initiate public health personell to strengthen clinical laboaratory diagnostic capacity and establish active surveillance program there by enable identifying and tracking bacterial outbreaks and severe diarrheal diseases. Ultimately, prevent occurrence of severe or fatal outbreak or diseases. We did a slight modification on it after your comment. 

19. The author should submit the structured questionnaire as the supplementary file which was used for the study.

• We took the comment and submited it. 

20. How Author has prevented the contamination of urine during stool sample collection?

21. Which type of sample container was used for the stool sample collection? Pl. describe. 

22. The author should briefly describe how aseptic conditions were maintained during stool sample collection.

• We considred the comments (20, 21 & 22) and we add further information on it. Training has been given to data collectors on the aseptic procedures of stool sample collection from children to avoid contamination of the stool samples (with urine, soil, or water). How to use the sterile collection materials (transport media, plastic wrap, and wooden sticks) and the critical steps in taking and transferring the stool samples were given during the training. 

23. A large number of typological errors were present throughout the manuscript. 

• We thank you for the comments and we considered it. We reviewed the errors throughout the manuscript. 

24. How stool samples were inoculated on MacConkey agar?

25. Stool samples were diluted or directly streaked on MacConkey agar?

26. The author should briefly describe the inoculation of stool samples on MacConkey agar.

• Based on the comments (24, 25, &26), we included some describtion. Faecal suspension was prepared by taking the wooded sticks (cotton swabs) from the transport media containing the stool sample and rinsed thoroughly in 1ml of saline. For the liquid stool sample, saline was not used. Then, a loopful of faecal suspension (liquid stool sample) was inoculated onto MacConkey agar. 

27. 16% glycerol stock was used?

28. Why author should store the E. coli isolates in Brain Heart Infusion agar? Generally, this agar is used for the cultivation of streptococci, meningococci, etc.

• Comment 27 & 28. Using broth (tryptic soy broth or brain heart infusion broth) with glycerol can be used for preservation of bacterial isolates. In our case, E. coli isolates from each cultured plate were stored at -80°C in brain heart infusion broth containing 16% (v/v) glycerol until use. The broth was used as storage media. 

29. Ampicillin concentration was 10mcg?

• Yes, it was ampicillin (10 µg)

30. Imipenem concentration was 30 mcg?

• No, it was meropenem (10 µg). And we made the correction. 

31. MDR was defined when the isolate was nonsusceptible to at least one agent among three or more antimicrobial categories. Is this the definition of MDR bacteria? The author should refer to the following paper for reference.

a. Singh AK, Das S, Kumar S, Gajamer VR, Najar IN, Lepcha YD, Tiwari HK, Singh S. Distribution of antibiotic-resistant Enterobacteriaceae pathogens in potable spring water of eastern Indian Himalayas: emphasis on virulence gene and antibiotic resistance genes in Escherichia coli. Frontiers in Microbiology. 2020 Nov 5;11:581072.

b. Singh AK, Das S, Singh S, Gajamer VR, Pradhan N, Lepcha YD, Tiwari HK. Prevalence of antibiotic resistance in commensal Escherichia coli among the children in rural hill communities of Northeast India. PloS one. 2018 Jun 18;13(6):e0199179.

• We thnak you for the concerns. The MDR definiation in our manuscript is based the paper: Magiorakos AP, Srinivasan A, Carey RB, Carmeli Y, Falagas ME, Giske CG, et al. Multidrug-resistant, extensively drug-resistant and pandrug-resistant bacteria: an international expert proposal for interim standard definitions for acquired resistance. Clin Microbiol Infect 2012;18:268–81. doi: 10.1111/j.1469-0691.2011.03570.x. The definition in this paper and in the one recommended above almost similar. In all case, MDR defined as resitance of antibiotics (at least one) in three or more classes. In our case: resistance (non-susceptible) to at least one agent (from each class) among three or more antimicrobial categories (classes) mean resitance of antibiotics in three or more classes. We put the reference while we defined it. And for better clarity we rewrote it: MDR was defined when the isolate was non-susceptible to at least one agent (from each class) in three or more antimicrobial classes.

32. What is the positive and negative control used for the Antibiotic Susceptibility test/

• E. coli ATCC 25922 was used as quality control strain (susceptible or positive control) for the antimicrobial susceptibility testing (as indicated in the CLSI guideline). And K. pneumoniae ATCC 700603 (positive control) and E. coli ATCC 25922 (negative control) for ESBL production, and K. pneumonia ATCC BAA-1705 (positive control) and K. pneumonia ATCC BAA-1706 (negative control) for carbapenemase production assay were used.

33. What are the inclusion and exclusion criteria used for the present study?

• Study participants were enrolled based on inclusion and exclusion criteria. The inclusion criteria were children under age five, who did not receive antibiotics in the past three weeks, and with diarrhea. Children with the age of five or older age, and who received antibiotics treatment were the exclusion criteria. The same inclusion criteria, except diarrheic, or those who attended the health facility for causes other than diarrhea were used to include non-diarrheic children in the study. 

34. What are the basic criteria used for the selection of antibiotics?

• Antibiotics for the study were selected based on CLSI guideline for enterobacterales. In the selection we also considered previous research and prescription trends in health facilities of Ethiopia. 

35. by using a 1μL size loop? What is the standard size of the inoculation loop? Is it 1 μL? 

• Thank you for the question. There are different sized inoculation loops including 1µl and 10µl inoculation loops. We used a 1µl inoculation loop based on the CLSI guideline recommendation. 

36. A confirmatory test for ESBLs activity was conducted for isolates resistant to cefotaxime (30μg) and ceftazidime (30μg) by the combination disk method. The author should briefly describe the screening test method and antibiotics used in the confirmatory test.

37. E. coli resistance to ceftazidime and cefotaxime were considered ESBL producers.

38. What is the role of clavulanic acid during ESBL detection?

39. Without using clavulanic acid author confirmed the ESBL pathogens.

• We thank you for the comment 36, 37, 38 and 39. We did it based on the CLSI guideline and the procedure explained in the guideline. Based on the comment we revised it. In the combination, clavulanic acid inhibit ESBL. DEC strains that were resistant at least to cefotaxime (30µg) or ceftazidime (30µg) in the screening test were selected. The ESBL production was confirmed using the combination disk method. Briefly, the combination disk method was done on MHA by using ceftazidime (CAZ) and cefotaxime (CTX) alone and with ceftazidime + clavulanic acid (CAZ/CLA) and cefotaxime + clavulanic acid (CTX/CLA) as recommended by CLSI 2020. The increase in zone size diameter by ≥5 mm for CTX/CLA and CAZ/CLA, when compared with that CTX and CAZ alone, was confirmed as the presence of ESBL.

40. Subsequently, the meropenem disks were removed from the broth using an inoculaton loop. Inoculaton????

• We corrected it: inoculation. 

41. DEC strains positive for the relevant virulence gene (s) stored in the Armauer Hansen Research Institute from previous works and CCUD (Culture Collection University of Gothenburg) 24T E. coli were used as the positive and negative control, respectively. What is 24T E. coli? The author should describe what is the positive and negative control.

• We used DEC pathotype strains which were whole genome sequenced and confirmed positive DEC strains (EPEC, ETEC, EIEC, EAEC, STEC, and DAEC) for the target virulence gene (s) as positive control. CCUD (Culture Collection University of Gothenburg) 24T E. coli which did not have the potential virulence factors of the pathogenic E. coli strains, and it was obtained from University of Gothenburg. We rewrote it. DEC pathotypes (whole genome sequenced strains) confirmed positive for the target virulence gene (s) from previous works used as a positive control. CCUD (Culture Collection University of Gothenburg) 24T E. coli strain confirmed negative to the target gene (s) was used as negative control.

42. An author should briefly and separately describe the number of samples collected both diarrheal as well as non-diarrheal during the dry season and rainy season.

• During the winter and spring (the dry season), 326 samples (273 diarrheic and 53 non-diarrheic children) were collected. The remaining 150 samples (118 diarrheic and 32 non-diarrheic children) were collected in the summer (the rainy season).

43. All 428 E. coli isolates were tested by PCR, of which 183 (42.8%, 183/428) were positive for DEC pathotype. The overall prevalence of DEC in the present study was 38.4% (183/476), and 58.5% (107/183) of the DEC types were isolated from Addis Ababa and the remaining from Debre Berhan. The line shows contradictory results to each other. The overall prevalence of DEC in the present study was 42.8% but again in the second line author mentioned that the overall prevalence of DEC type is 38.4%. The author should briefly revise the sentence.

• The total samples included in the study was 476, of which we got 183 DEC strains. That mean, the prevalence of DEC is 183/476 (38.4%). 42.8% is calculated from total E. coli isolates (428), 183/428 and it is not from the total study participants. After considering your comment, we amended the statement by deleting the the figure, (42.8%, 183/428), only 183 used to avoid the confusion. All 428 E. coli isolates were tested by PCR, of which 183 were positive for DEC pathotype.

44. The overall prevalence of DEC in the present study was 38.4% (183/476), and 58.5% (107/183) of the DEC types were isolated from Addis Ababa and the remaining from Debre Berhan (Table 3, Fig1 a & b). The current result and table 3 data were not clear. 

• We revised it. The overall prevalence of DEC in the present study was 38.4% (183/476). Of the total DEC pathotypes, 58.5% (107/183) were from Addis Ababa, and the remaining 41.5% (76/183) were from Debre Berhan (Table 3, Fig1 a & b). The prevalence of DEC during the winter and spring (the dry season) and the summer (the rainy season) was 38% (124/326) and 39.3% (59/150), respectively. 

45. In Table 3 what are the total cases meaning? And what is total control?

• Total case and total control were to mean total diarrheic and total non-diarrheic regardless of the study areas, respectively. We revised it now, in place of total case and total control, we used total diarrheic and total non-diarrheic, respectively. 

46. Dry season? What is the meaning of the dry season? The author did not mention any data on the dry season in table 2.

• In the present study, winter, spring and summer were included. Winter and spring are dry season whereas summer is a rainy season. We indicated it in bracket in the first use, e.g. winter and spring (dry season) to mean dry season in the present study include winter and summer. We did a slight revision based on the comments. 

47. Under-five children were more likely to acquire DEC during the rainy season compared to the dry season (AOR=0.529, CI= 0.335; 0.835) (Table 2). In table 2 author mentioned the data of summer, Winter, and Spring whereas in the manuscript it was written as dry season and rainy season as well as some places' rainy season. The author should write it properly in the either rainy and dry seasons and spring, summer, and winter in tabl2 as well as the manuscript write-up.

• We took the comment and made a revision. We used winter and summer. 

48. The data of table 2 shows that the most prevalent season for DEC was winter (45.9%) followed by summer (32.2%) and spring (21.9%). Kindly check it.

• In the table 2, the data indicated the proportion not the prevalence. This mean, of 183 DEC, 84 were in winter (84/183=45.9%), 40 in spring (40/183=21.9%), and 59 in summer (59/183=32.2%). Prevalence=DEC positive/total children under age five visit during that season. We thank you for your comments. 

49. Resistance to ceftazidime and cefotaxime was higher in DEC isolates obtained from diarrheic compared to non-diarrheic children (p=0.010). All the STEC and DAEC, 15% ETEC (6/39), and 22% EAEC (17/76) were resistant to ciprofloxacin. Kindly rephrase the sentence properly.

• DEC strains resistant to ceftazidime and cefotaxime were higher in diarrheic compared to non-diarrheic children (p=0.010). All the three STECs and 1 DAEC, 15% ETEC (6/39), and 22% EAEC (17/76) strains were resistant to ciprofloxacin.

50. ETEC (5%, 2/39) and EAEC (3%, 2/76) were resistant to meropenem and ertapenem. The author should separately describe what percentage of ETEC as well as EAEC were resistant to meropenem as well as entrapenem. 

• We took the comment. Two ETECs (5%, 2/39) and two EAECs (3%, 2/76) were resistant to both meropenem and ertapenem.

51. This finding agrees with reports from Nigeria [42], and Iran [40]. The author should add the percentage DEC in diarrheic and non-diarrheic in Nigeria and Iran.

• We took the comment and revised it. A higher prevalence of DEC (40.7%) was detected in diarrheic compared to non-diarrheic children under age five (28.2%) in the present study (p- 0.020). This finding agrees with reports from Nigeria, 76.7% and 54.6%, and Iran, 90% and 20%, in diarrheic and non-diarrheic, respectively.

52. Only LT-positive ETEC was found in the non-diarrheic children in the present study. ST-positive ETEC strains are commonly associated with diarrhea compared to LT-positive. LT toxin is not responsible for diarrhea?

• It is responsible. It was to mean that the ST-positive ETEC is more frequently found in severe infections compared to LT-positive. Revised accordingly. 

53. EAEC (41.5%) was the predominant DEC pathotype identified in the present study. What could be the plausible region?

• The possible reason for its predominant epidemiology may be associated with its genomic plasticity. It is relatively more genetically hetrogenetic and an emerging pathogen. The heterogeneity of EAEC could contribute for its epidemiology. 

54. The second most prevalent DEC pathotype was ETEC (21.3%) which was consistent with studies done in Nigeria [42], China [21], Sudan [38], and Iran [40]. The author should kindly add the data of these countries.

• We did the revision. The second most prevalent DEC pathotype was ETEC (21.3%) which was consistent with reports of 17.3% in Nigeria [42], 14.8% in China [21], 18% in Sudan [38], and 26% in Iran [40].

55. In table 2, the acquisition of DEC in the rainy season (32.2%) looks less than in the dry season (45.9%). It is quite confusing some tables show data for winter, spring, and summer whereas the author discusses the dry season and rainy season. The author should kindly revise it thoroughly and use either winter, summer as well as spring or dry and rainy seasons. What could be the possible reason for the prevalence of DEC pathotypes in winter as compared to the rainy season? The author should discuss it with suitable reason.

• We took the comments to avoid the confusion. And we used winter (dry season) and summer (rainy season). 32.2% for summer (rainy season) is not a prevalence, it is a proportion (59/183, not 59/150). Actual prevalence for summer is 59/150=39.3% and for winter 84/230=36.5%. DEC is more prevalence in summer than winter and we reviewed it. 

56. In table 2, the acquisition of DEC in the rainy season (32.2%) looks less than in the dry season (45.9%). However, it was statistically significant that under-five children were more likely to acquire DEC during the rainy season compared to the dry season in the present study, resulting in children visiting the health facility in the summer due to DEC infections. If children have visited the health facility center in the rainy season then why is the prevalence of DEC pathotypes shown higher in the winter season?

• See response to comment 55. In fact, the prevalence is higher in summer compared to winter in this case. 

57. Children who were cared for by their mothers (37.5%) had statistically significant protection than those who were cared for by others (grandmothers, close family members, and daycare) (14.0%). The author should recheck the data properly. It is 37.5% or 53.6%??

• The figures are correct, we took the DEC negative in both cases. In table 2, when we look the proportion for DEC positive in children cared by mother (53.6%) is higher than others (8.2%). If we calculate the prevalence of DEC in children cared by mother (98/208=47%) and in other (15/56=27%). Still the prevalence is higher in children cared by mother than cared by other. But, this is not ststistically significant. Instead, the statistical value indicated that children who cared by mother are protected compared to children cared by other. We also tried to see the actual occurrence of an event (DEC) through odd value. Odd is the probability of an event occurring divided by the probability of an event not occurring. The higher odd value for an event, is the less likely to occur (vice versa). The odd for DEC positive with mothers is (98/110=0.891); the odd for DEC positive with other is (15/41=0.366). The lower the odd is more likely to occur. In this case, DEC more occur with other (regardless of the proportion and prevalence value). Thus, in the discussion, we used the figure of DEC negative to avoid confusion. Children who were cared for by their mothers had statistically significant protection (37.5% DEC negative) than those who were cared for by others (grandmothers, close family members, and daycare) (14.0% DEC negative).

58. This may be due to that the mother is more likely careful than other caregivers, potentially contributing to the incidence of DEC in the children. What could be the possible reason for the prevalence of DEC pathotypes in more who cares by mother as compared to other caregivers? 

• See response to 57

59. Any significant effect of the number of children in a particular family on the prevalence of DEC pathotype in that particular family?

• Number of children in the family did not found associated with DEC acquisition in the present study (p>0.25) and not included in the multivariate logistic regression.

60. What is the meaning of total control which showed in table 3?

• Based on your comment, we revised it (see response to comment 45). 

61. All 428 E. coli isolates were tested by PCR, of which 183 (42.8%, 183/428) were positive for DEC pathotype. The overall prevalence of DEC in the present study was 38.4% (183/476), and 58.5% (107/183) of the DEC types were isolated from Addis Ababa and the remaining from Debre Berhan (Table 3, Fig1 a & b). The author should kindly revise it properly. It creates confusion like 42.8%% were DEC type and the next line overall prevalence was 38.4%. 

• 183 is from 428 total E. coli (DEC from total E.coli isolates, not from total study participants). The 428 E.coli were from 476 children. The total sample size is 476. And the prevalence of DEC among the 476 children is 183/476=38.4%. And we put the numerator and denomnnator to avoid the confusion. 

62. In table 3 the DAEC prevalence was showing 14.5% whereas in the manuscript it was written as 41.5%. The author should revise the tables and write either in percentages or numbers.

• In the present study, among the pathotypes, DAEC (0.6%, 1/183) and EAEC (41.5%, 76/183) were detected. We checked it. 

63. The author should revise the manuscript thoroughly and correct the data in the table as well as the manuscript. 

• We took all the comments in to consideration and did the revision. Lastly, we provide our gratitude for your constructive comments.

---

## [Decision Letter · Decision Letter 3]

17 May 2023

PONE-D-22-27001R3Molecular epidemiology and antimicrobial susceptibility of diarrheagenic Escherichia coli isolated from children under age five with and without diarrhea in Central EthiopiaPLOS ONE

Dear Dr. Zelelie,

Thank you for submitting your manuscript to PLOS ONE. After careful consideration, we feel that it has merit but does not fully meet PLOS ONE’s publication criteria as it currently stands. Therefore, we invite you to submit a revised version of the manuscript that addresses the points raised during the review process.

We look forward to receiving your revised manuscript.

Kind regards,

Samer Singh, Ph.D.

Academic Editor

PLOS ONE

Journal Requirements:

Additional Editor Comments:

There are some issue with the language at few places as indicated by the reviewer. The changes suggested are indicative. Authors would like to check the manuscript thoroughly for typographical and grammatical errors and suitably correct them as needed. 

Reviewers' comments:

Reviewer's Responses to Questions

**Comments to the Author**

1. If the authors have adequately addressed your comments raised in a previous round of review and you feel that this manuscript is now acceptable for publication, you may indicate that here to bypass the “Comments to the Author” section, enter your conflict of interest statement in the “Confidential to Editor” section, and submit your "Accept" recommendation.

Reviewer #3: All comments have been addressed

2. Is the manuscript technically sound, and do the data support the conclusions?

Reviewer #3: Yes

3. Has the statistical analysis been performed appropriately and rigorously? 

Reviewer #3: Yes

4. Have the authors made all data underlying the findings in their manuscript fully available?

Reviewer #3: Yes

5. Is the manuscript presented in an intelligible fashion and written in standard English?

Reviewer #3: Yes

6. Review Comments to the Author

Reviewer #3: Q.1 Then, the disks were placed on Mueller-Hinton agar plates (Oxoid, UK) freshly inoculated with a 0.5 McFarland suspension of Escherichia coli ATCC 25922. The results from the ESBL confirmation and carbapenemase production were interpreted according to CLSI guidelines.

Kindly italicized the coli also.

Q.2 DEC pathotypes (whole genome sequenced strains) confirmed positive for the target virulence gene(s) from previous works used as a positive control. CCUD (Culture Collection University of Gothenburg) 24T E. coli strain confirmed negative to the target gene (s) was used as a negative control. Kindly correct the typological errors.

Q.3 A clinical presentation of children under age five positive for DEC is presented in S2Table. Pl. correct the typological errors.

Q.4 Of the total DEC pathotypes, 58.5% (107/183) were from Addis Ababa, and the remaining 41.5% (76/183) were from Debre Berhan (Table 3, Fig1 a & b). Table 3 or Table 2?? Pl. correct it carefully.

Q.4 Placed in a freezer at −20°C for 10 minutes. Kindly correct the typological errors.

Q.5 Then, 100μL of the supernatant was transferred into a nuclease-free Eppendorf tube and stored at −20°C until use. Kindly correct the typological errors.

Q.6 PCR reaction 3 contained primer mix 3 (M-3) for the detection of EAEC and DAEC targeting virulence genes aggR, astA, and daaF. Why italicized the 3?

Q.7 The PCR thermal conditions were set with an initial denaturation of 94°C for 2 min, followed by 35 cycles of 92°C for 30 sec, annealing at 60°C for 30 sec, extension at 72°C for 30 sec, and a final extension at 72°C for 5 min in a PCR machine (Biometra TRIO Thermal Cycler, Analytik Jena). Kindly correct the typological errors.

Q.8 Table 3 and Table 4 were showing the number of isolates as well as the percentage also. It creates confusion for the readers. Kindly show the data only in percentages is better to understand.

Q.9 Author is showing two types of data one is the proportion of DEC pathotypes and the second one is the prevalence of DEC pathotypes. It will create confusion for the readers.

Q.10 What could be the plausible reason to collect the diarrheal stool sample from the children who did not receive the antibiotics in the past three weeks?

Q.11 Table 4. Antimicrobial resistant profile of DEC pathotypes (n=183) isolated from diarrheic and non-diarrheic under -five children, Addis Ababa and Debre Berhan, Ethiopia. Kindly correct it properly.

7. PLOS authors have the option to publish the peer review history of their article (what does this mean?). If published, this will include your full peer review and any attached files.

Reviewer #3: **Yes**

---

## [Author Response · Author response to Decision Letter 3]

2 Jun 2023

For all academic editor and reviewer(s):

First, we thank you for the comments. We have learnt a lot from your comments and your comments make our manuscript more important than it was. In addition, during the review process we got opportunity to improve the manuscript. Your additional comment further improves the manuscript. And for this we kindly provide our gratitude to you for your professional supports. We thank you to each comments and concerns. Below we go points to points. 

Minor comments

Q.1 Then, the disks were placed on Mueller-Hinton agar plates (Oxoid, UK) freshly inoculated with a 0.5 McFarland suspension of Escherichia coli ATCC 25922. The results from the ESBL confirmation and carbapenemase production were interpreted according to CLSI guidelines.

Kindly italicized the coli also.

• We thank you for the comment. We italiced it. 

Q.2 DEC pathotypes (whole genome sequenced strains) confirmed positive for the target virulence gene(s) from previous works used as a positive control. CCUD (Culture Collection University of Gothenburg) 24T E. coli strain confirmed negative to the target gene (s) was used as a negative control. Kindly correct the typological errors.

• We corrected it. DEC pathotypes (whole genome sequenced strains) confirmed positive for the target virulence gene(s) from previous works used as the positive control. The CCUG (Culture Collection University of Gothenburg) 24T E. coli strain confirmed negative for the target gene (s) was used as a negative control.

Q.3 A clinical presentation of children under age five positive for DEC is presented in S2Table. Pl. correct the typological errors.

• We corrected it. The clinical presentation of children under age five positive for DEC is presented in S2 Table.

Q.4 Of the total DEC pathotypes, 58.5% (107/183) were from Addis Ababa, and the remaining 41.5% (76/183) were from Debre Berhan (Table 3, Fig1 a & b). Table 3 or Table 2?? Pl. correct it carefully.

• We corrected it. It is Table 2. (table 3 also have the data but table 2 is more clear). 

Q.4 Placed in a freezer at −20°C for 10 minutes. Kindly correct the typological errors.

• We corrected it. −20 °C

Q.5 Then, 100μL of the supernatant was transferred into a nuclease-free Eppendorf tube and stored at −20°C until use. Kindly correct the typological errors.

• We corrected it. −20 °C

Q.6 PCR reaction 3 contained primer mix 3 (M-3) for the detection of EAEC and DAEC targeting virulence genes aggR, astA, and daaF. Why italicized the 3? In case it has special meaning, state it, or correct it.

• aggR, astA, and daaF are among the target genes and we italized all the genes (like bfp, eae, and stx as well as lt, st, virF, and ipaH). 

Q.7 The PCR thermal conditions were set with an initial denaturation of 94°C for 2 min, followed by 35 cycles of 92°C for 30 sec, annealing at 60°C for 30 sec, extension at 72°C for 30 sec, and a final extension at 72°C for 5 min in a PCR machine (Biometra TRIO Thermal Cycler, Analytik Jena). Kindly correct the typological errors.

• We corrected it. The PCR thermal conditions were set with an initial denaturation of 94 °C for 2 min, followed by 35 cycles of 92 °C for 30 sec, annealing at 60 °C for 30 sec, extension at 72 °C for 30 sec, and a final extension at 72 °C for 5 min in a PCR machine (Biometra TRIO Thermal Cycler, Analytik Jena).

Q.8 Table 3 and Table 4 were showing the number of isolates as well as the percentage also. It creates confusion for the readers. Kindly show the data only in percentages as it is better to understand.

• Based on the comment, we put the data in percentage for both tables. 

Q.9 Author is showing two types of data one is the proportion of DEC pathotypes and the second one is the prevalence of DEC pathotypes. It will create confusion for the readers. Suitably change to make them comparable.

• Thank you for the concerns. We use proportions in all our tables. The proportion could indicate the DEC positive from the total for each variable (e.g. in table 2). We use prevalence in some cases (data in table 2) in the discusions to make more clear on the issue (to avoid confusion for the readers). 

Q.10 What could be the plausible reason to collect the diarrheal stool sample from the children who did not receive the antibiotics in the past three weeks? Briefly indicate reason/logic.

• Prior antibiotic use can affect pathogen distribution [33]. Although the exact time for antibiotics to clear from the body is varied (due to many factors), some antibiotics take a short time (24hrs) [34], and others, such as azithromycin [35], take a longer time (more than two weeks). For this reason, only those who had no antibiotic exposure in the past three weeks were included in the study. 

Q.11 Table 4. Antimicrobial resistant profile of DEC pathotypes (n=183) isolated from diarrheic and non-diarrheic under -five children, Addis Ababa and Debre Berhan, Ethiopia. Kindly correct it properly.

• We corrected it. Antimicrobial resistant profile of DEC pathotypes (n=183) isolated from diarrheic and non-diarrheic children under age five in Addis Ababa and Debre Berhan, Ethiopia.

---

## [Decision Letter · Decision Letter 4]

29 Jun 2023

Molecular epidemiology and antimicrobial susceptibility of diarrheagenic Escherichia coli isolated from children under age five with and without diarrhea in Central Ethiopia

PONE-D-22-27001R4

Dear Dr. Zelelie,

We’re pleased to inform you that your manuscript has been judged scientifically suitable for publication and will be formally accepted for publication once it meets all outstanding technical requirements.

Kind regards,

Samer Singh, Ph.D.

Academic Editor

PLOS ONE

Additional Editor Comments (optional):

Reviewers' comments:

Reviewer's Responses to Questions

**Comments to the Author**

1. If the authors have adequately addressed your comments raised in a previous round of review and you feel that this manuscript is now acceptable for publication, you may indicate that here to bypass the “Comments to the Author” section, enter your conflict of interest statement in the “Confidential to Editor” section, and submit your "Accept" recommendation.

Reviewer #3: (No Response)

2. Is the manuscript technically sound, and do the data support the conclusions?

Reviewer #3: Yes

3. Has the statistical analysis been performed appropriately and rigorously? 

Reviewer #3: Yes

4. Have the authors made all data underlying the findings in their manuscript fully available?

Reviewer #3: Yes

5. Is the manuscript presented in an intelligible fashion and written in standard English?

Reviewer #3: Yes

6. Review Comments to the Author

Reviewer #3: (No Response)

7. PLOS authors have the option to publish the peer review history of their article (what does this mean?). If published, this will include your full peer review and any attached files.

Reviewer #3: **Yes: **Dr. Ashish Kumar Singh, Ph.D.

---

## [Editor Report · Acceptance letter]

7 Jul 2023

PONE-D-22-27001R4 

Molecular epidemiology and antimicrobial susceptibility of diarrheagenic *Escherichia coli* isolated from children under age five with and without diarrhea in Central Ethiopia   

Dear Dr. Zelelie:

I'm pleased to inform you that your manuscript has been deemed suitable for publication in PLOS ONE. Congratulations! Your manuscript is now with our production department. 

Kind regards, 

on behalf of

Dr Samer Singh 

Academic Editor

PLOS ONE